# Cyanobacterial antimetabolite 7-deoxy-sedoheptulose blocks the shikimate pathway to inhibit the growth of prototrophic organisms

Klaus Brilisauer [1,2], Johanna Rapp [2], Pascal Rath[1], Anna Schöllhorn [2], Lisa Bleul[3], Elisabeth Weiß[3], Mark Stahl[4], Stephanie Grond[1] & Karl Forchhammer [2]

Antimetabolites are small molecules that inhibit enzymes by mimicking physiological substrates. We report the discovery and structural elucidation of the antimetabolite 7-deoxy-sedoheptulose (7dSh). This unusual sugar inhibits the growth of various prototrophic organisms, including species of cyanobacteria, *Saccharomyces*, and *Arabidopsis*. We isolate bioactive 7dSh from culture supernatants of the cyanobacterium *Synechococcus elongatus*. A chemoenzymatic synthesis of 7dSh using *S. elongatus* transketolase as catalyst and 5-deoxy-D-ribose as substrate allows antimicrobial and herbicidal bioprofiling. Organisms treated with 7dSh accumulate 3-deoxy-D-*arabino*-heptulosonate 7-phosphate, which indicates that the molecular target is 3-dehydroquinate synthase, a key enzyme of the shikimate pathway, which is absent in humans and animals. The herbicidal activity of 7dSh is in the low micromolar range. No cytotoxic effects on mammalian cells have been observed. We propose that the in vivo inhibition of the shikimate pathway makes 7dSh a natural antimicrobial and herbicidal agent.

[1] Institute of Organic Chemistry, Eberhard Karls Universität Tübingen, Auf der Morgenstelle 18, 72076 Tübingen, Germany. [2] Microbiology, Organismic Interactions, Eberhard Karls Universität Tübingen, Auf der Morgenstelle 28, 72076 Tübingen, Germany. [3] Interfaculty Institute of Microbiology and Infection Medicine Tübingen (IMIT), Eberhard Karls Universität Tübingen, Eugenstraße 6, 72076 Tübingen, Germany. [4] Center for Plant Molecular Biology, Eberhard Karls Universität Tübingen, Auf der Morgenstelle 32, 72076 Tübingen, Germany. Correspondence and requests for materials should be addressed to S.G. (email: biomolchemie@orgchem.uni-tuebingen.de) or to K.F. (email: biomolchemie@orgchem.uni-tuebingen.de)

Cyanobacteria—the dominating photoautotrophic, oxygen-producing microbes on earth—have gained increasing attention in natural product research in recent years. Their omnipresence in the light-exposed biosphere is based on a large repertoire of survival strategies for withstanding challenging environmental conditions and protecting their niches against competitors. To this end, cyanobacteria produce a wide range of secondary metabolites, often with a unique composition and specialized functions, which mediate manifold processes, such as chemical defense[1], preservation[2], and quorum sensing[3]. Some of the known cyanobacterial metabolites exhibit antiviral[4], antibacterial[5], antifungal[6], or herbicidal[7] activities, with promising possible applications in human health, agriculture, or industry[8].

Although cyanobacteria produce a broad range of bioactive compounds in terms of structure and targets, only few are described as classical antimetabolites. Antimetabolites are chemical analogs of the natural substrates of enzymes, where they bind to the active site but are not converted to the functional product. In this way, antimetabolites block a biological process, such as a biosynthetic pathway. One of the best-studied cyanobacterial antimetabolites is the non-proteinogenic amino acid β-methylamino-L-alanine (BMAA), which was initially isolated from cultures of species of Nostoc[9]. BMAA can be mistakenly incorporated into nascent proteins in place of L-serine, which possibly causes protein misfolding and aggregation[10].

Antimetabolites are useful for controlling the growth of microorganisms, fungi, and plants. To avoid harmful side effects, these compounds should target biological processes that do not occur in animals, especially mammals. One such process involves the enzymes of the shikimate pathway, in which seven enzymes catalyze the sequential conversion of erythrose 4-phosphate and phosphoenolpyruvate (PEP) via shikimate to chorismate[11], the essential precursor of the aromatic amino acids phenylalanine, tyrosine, and tryptophan. Each enzyme of the shikimate pathway catalyzes an essential reaction in chorismate biosynthesis that cannot be bypassed by an alternative enzyme. The inhibition of any enzyme of this pathway, therefore, leads to impairment of the entire cell metabolism and results in arrested growth or even cell death[12]. Chemical compounds that interfere specifically with any enzyme activity in this pathway are considered harmless for humans and other mammals when handled at reasonable concentrations[13]. Therefore, the enzymes of the shikimate pathway are attractive potential targets for the development of novel antimetabolites.

One of the most prominent antimetabolites that targets the shikimate pathway is the synthetic herbicide glyphosate [N-(phosphonomethyl)glycine][14], whose use is intensely discussed, most recently due to its perturbation of the gut microbiota of honey bees[15]. Since its initial commercialization in 1974, glyphosate has become the main component of various total herbicides applied in agriculture, industry and private households, today in amounts of >800,000 tons per year[16]. Glyphosate is a potent inhibitor of the 5-enolpyruvylshikimate 3-phosphate (EPSP) synthase, which converts PEP and shikimate 3-phosphate to EPSP. As a transition state analog of PEP, glyphosate leads to the accumulation of shikimate 3-phosphate[17]. This blocking of the synthesis of the end products of the aromatic pathway results in perturbation of metabolic homeostasis and eventually leads to cell death. To our knowledge, glyphosate is the only bioactive compound with a potent in vivo inhibitory effect on the shikimate pathway that has been described to date. Another target of antimetabolites is the first enzyme in branched-chain amino acid synthesis, acetohydroxyacid synthase (AHAS, E.C. 2.2.1.6). Like the shikimate pathway, the branched-chain amino acid synthesis pathway is only found in plants, bacteria and fungi and therefore, compounds inhibiting the AHAS are highly successful commercial herbicides[18,19].

Allelochemicals, bioactive compounds for the inhibition of rival organisms, play a major role in cyanobacterial niche competition. Both filamentous and colonial cyanobacteria are known to be potent producers of a wide variety of allelochemicals and other secondary metabolites[20,21]. By contrast, little is known about the synthesis of such metabolites by simple unicellular cyanobacteria like Synechococcus and Prochlorococcus species. Indeed, for a long time it was not even anticipated that these cyanobacteria produce bioactive metabolites because of their small, stream-lined genomes and lack of non-ribosomal peptide synthase gene clusters[22]. However, newer findings suggest an extensive ability of simple unicellular cyanobacteria for the production of secondary metabolites, which is mainly based on catalytic promiscuity[23].

Synechococcus elongatus PCC 7942 is one of the most commonly used model organisms for molecular genetic studies in cyanobacteria[24]. Its circular chromosome (ca. 2.7 Mb, GenBank accession no. CP000100) and plasmids (GenBank accession nos. AF441790 and S89470) lack apparent gene clusters for the synthesis of complex secondary metabolites[25]. However, it has been reported that collapsing aged cultures of S. elongatus secrete a non-identified hydrophobic metabolite that inhibits the growth of a large variety of photosynthetic organisms[26].

In this work, we identify an anti-cyanobacterial bioactivity in supernatants of stationary S. elongatus cultures. We assign this bioactivity to a hydrophilic compound that therefore differs from the metabolite cited above. Subsequent bioactivity-guided isolation, structural elucidation, and characterization of the mode of action reveal the first identified natural antimetabolite that targets the shikimate pathway in vivo.

## Results

**Isolation of the bioactive metabolite.** Supernatants of stationary cultures of S. elongatus inhibited the growth of Anabaena variabilis. The inhibitory activity could be extracted from lyophilized culture supernatants with the polar solvent methanol, but not with chloroform, acetone, or ethyl acetate as visualized by agar-diffusion plate assays (Fig. 1a). The producer strain was not affected by these extracts. Significant production of the inhibitor required $CO_2$ supplementation of liquid cultures and was dependent on the cell density of the producer strain. Inhibitor content apparently peaked after about 2 weeks of growth of S. elongatus cultivated in batch cultures in BG11 medium (Fig. 1b).

The chemical characterization of the bioactive compound indicated high polarity and absence of UV absorption. The low levels produced demanded an optimized bioactivity-guided isolation protocol with several enrichment and purification steps. A chromatographically pure compound was obtained via successive size-exclusion chromatography, medium-pressure liquid chromatography (MPLC) on normal phase, and ligand/ion-exchange high-performance liquid chromatography (HPLC) coupled to evaporative light-scattering detection (ELSD) (Supplementary Fig. 1). The molecular formula of the bioactive molecule was determined by electrospray ionization high-resolution mass spectrometry (ESI-HRMS) to be $C_7H_{14}O_6$ ($M_R = 194.18$ Da from $m/z = 217.0675$ $[M+Na]^+$) (Supplementary Fig. 2).

We elucidated the structure of the chromatographically pure compound using nuclear magnetic resonance (¹H-NMR, ¹³C-NMR, and two-dimensional spectra; Supplementary Table 1, Supplementary Figs. 12–16). The signals were assigned to the constitution of a 7-deoxyheptulose and indicated the relative

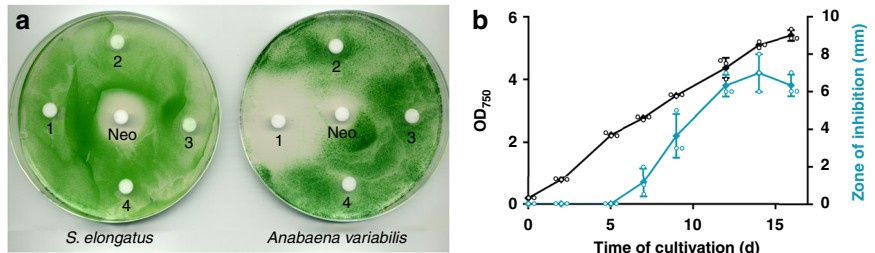

**Fig. 1** Extracts of supernatant of *Synechococcus elongatus* inhibits growth of *Anabaena variabilis*. **a** Agar-diffusion plate assay with the effect of organic extracts (1, methanol; 2, chloroform; 3, acetone, and 4, ethyl acetate) of lyophilized supernatant of stationary-phase *S. elongatus* cultures on the growth of the producer strain and *A. variabilis*. Neomycin (Neo, 20 μg) served as positive control. **b** Optical density of producer strain *S. elongatus* (black) and zone of *A. variabilis* growth inhibition (diameter) of methanol extracts of *S. elongatus* supernatant on agar diffusion plates (turquoise). Values represent the mean values of three biological replicates; standard deviations are indicated. Dots indicate data distribution. Source data are provided as a Source Data file

**Fig. 2** Structure and chemoenzymatic synthesis of 7-deoxy-sedoheptulose (7dSh, **1**). **a** Chemoenzymatic synthesis of 7-deoxy-sedoheptulose (7dSh). Absolute configurations of stereo-centers are indicated. **b** Chemical structure of 7dSh in the furanose form with given assignments of coupling constants (gray). **c** (1–4) $^1$H NMR spectra of 7dSh (CD$_3$OD, 600 MHz) chromatographically purified from supernatants of stationary phase cultures of *S. elongatus* (1, green), of the purified 7dSh from the supernatants of *Streptomyces setonensis* as control (2, red), and of enzymatically synthesized 7dSh (3, black). Predicted from assigned NMR-data (4, blue) of 7dSh in the 7-deoxy-D-*altro*-heptulofuranose form (Bruker, TopSpin software). Additional proton NMR signals in 1–3 give evidence for the dynamic forms of 7dSh in solution (open chain tautomers, ring conformers)

configuration mainly present in the furanose form (Fig. 2b, c). In marked contrast to six-membered sugar rings (pyranoses), the five-membered furanoses exhibit complex multiple ring conformations, and the coupling pattern only allows a suggested

relative configuration[27]. The occurrence of pyranose and furanose forms of D-2-heptuloses are known for D-*altro*-2-heptulose, D-*manno*-2-heptulose, D-*galacto*-2-heptulose, and D-*gluco*-2-heptulose[28]. The only D-2-heptulose existing mainly in furanose form

corresponds to the *altro* configuration, which rendered this configuration most probable for the inhibitor isolated from culture supernatants of *S. elongatus*.

**Chemoenzymatic synthesis of 7-deoxy-sedoheptulose**. To unambiguously prove the chemical structure of the 7-deoxyheptulose from *S. elongatus* culture supernatants, we established the chemoenzymatic synthesis of 7-deoxy-D-*altro*-2-heptulose (**1**) (7-deoxy-sedoheptulose, 7dSh). $C_7$-carbohydrate intermediates occur in the pentose phosphate pathway and can be biosynthesized by the transfer of a $C_2$-unit onto a $C_5$-precursor using the enzyme transketolase. Transketolase (EC 2.2.1.1) stereospecifically adds the nucleophile to the *re*-face of the D-enantiomers of 2-hydroxyaldehydes (aldoses) and controls the stereochemistry of the reaction to result in (3S, 4R)-configured ketoses[29]. We cloned the gene encoding the *S. elongatus* transketolase (Synpcc7942_0538) in an *Escherichia coli* His-tag (pET15b) overexpression vector and purified the recombinant protein by affinity chromatography (see Methods). In the enzymatic synthesis of 7dSh, recombinant *S. elongatus* transketolase transfers the C1–C2 ketol unit of β-hydroxypyruvate (**3**) to 5-deoxy-D-ribose (**2**) in the presence of thiamine diphosphate and divalent cations ($Mg^{2+}$)[30] (Fig. 2a). Release of $CO_2$ from β-hydroxypyruvate during the transketolase reaction prevents the back-reaction and enables a one-way synthesis of 7-deoxy-D-*altro*-2-heptulose (**1**), which is the only product according to NMR and MS data.

Transketolases efficiently react with phosphorylated sugars, but reactions with dephosphorylated sugars result in low yields[31]. In agreement, the chemoenzymatic synthesis of 7dSh from 5-deoxy-D-ribose gave yields of about 20%. We purified chemoenzymatically synthesized 7dSh following the same protocol used for purifying 7dSh from culture supernatants (Supplementary Fig. 1), except that size-exclusion chromatography on Sephadex LH20 could be omitted.

The $^1$H-NMR spectrum of chemoenzymatically synthesized 7dSh (**1**) was identical to that of the compound isolated from *S. elongatus* culture supernatant. The chemical structure of 7dSh was reported in 1970 as the metabolite SF-666B from *Streptomyces setonensis* nav. sp. by Ezaki, Tsuruoka[32]. SF-666B was described to show exclusive activity against *Gluconobacter oxydans* subsp. *suboxydans* at low micromolar concentrations ($0.8\ \mu g\ mL^{-1}$)[33]. Therefore, we isolated SF-666B from culture supernatants of the *Streptomyces setonensis* production strain

following our purification protocol (Supplementary Fig. 1). NMR spectroscopy revealed that SF-666B is indeed identical to 7dSh isolated from *S. elongatus* culture supernatants and to chemoenzymatically synthesized 7dSh (Fig. 2c).

**Activity of 7dSh against cyanobacterial strains**. With the assigned structure of 7dSh (**1**) and milligram amounts of pure compound at hand, we aimed for detailed biological profiling of the compound. In contrast to the previously reported activity of SF-666B, none of the 7dSh preparations (chemoenzymatically synthesized, purified from culture supernatants of *S. elongatus* or *Streptomyces setonensis*) showed any activity against *Gluconobacter oxydans* under the previously described assay conditions[33] and under various other tested conditions (not shown). By contrast, all 7dSh preparations inhibited the growth of the filamentous cyanobacterium *A. variabilis*. To clarify the biological activity of 7dSh and its biological mode of action, we first analyzed the effects of 7dSh on cyanobacteria in more detail (Fig. 3).

The effect of 7dSh (**1**) on *A. variabilis* depended on the ratio between the 7dSh concentration and the cell density of the cultures (determined as optical density at 750 nm, $OD_{750}$; Fig. 3a). Cultures with an initial $OD_{750}$ of 0.5 were hardly affected by 7dSh concentrations up to $5\ \mu g\ mL^{-1}$ (ca. 25 μM). When the cell density of *A. variabilis* was lowered to an $OD_{750}$ of 0.2, 7dSh had a dose-dependent effect. At a concentration of $2.5\ \mu g\ mL^{-1}$ (ca. 13 μM), 7dSh showed a cytostatic effect. A further increase of 7dSh to $5\ \mu g\ mL^{-1}$ resulted in lysis of the cells. With even lower initial cell densities (initial $OD_{750} < 0.05$), the effect of 7dSh was even more pronounced; already $2.5\ \mu g\ mL^{-1}$ 7dSh had a bactericidal effect. Therefore, the effect of 7dSh on *A. variabilis* can be either bacteriostatic or bactericidal, depending on the amount of 7dSh available per cell. This result indicated a cellular binding site for 7dSh that reduces the titer of the compound in solution or a metabolic alteration of 7dSh.

We subsequently used bactericidal concentrations of 7dSh (**1**) for bioprofiling to obtain unambiguous results. Since 7dSh was active against a cyanobacterium but not against *Gluconobacter oxydans*, we speculated that 7dSh might target the photosynthetic apparatus. Treatment of *A. variabilis* with 7dSh (ca. 50 μM) led to a slow decrease in photosynthetic oxygen formation over a period of 24 h (Fig. 3b). This effect is in contrast to that of specific inhibitors of photosynthesis, such as 3-(3,4-dichlorophenyl)-1,1-dimethylurea, which act almost immediately. The slow decrease resembled the effect of the protein synthesis inhibitor

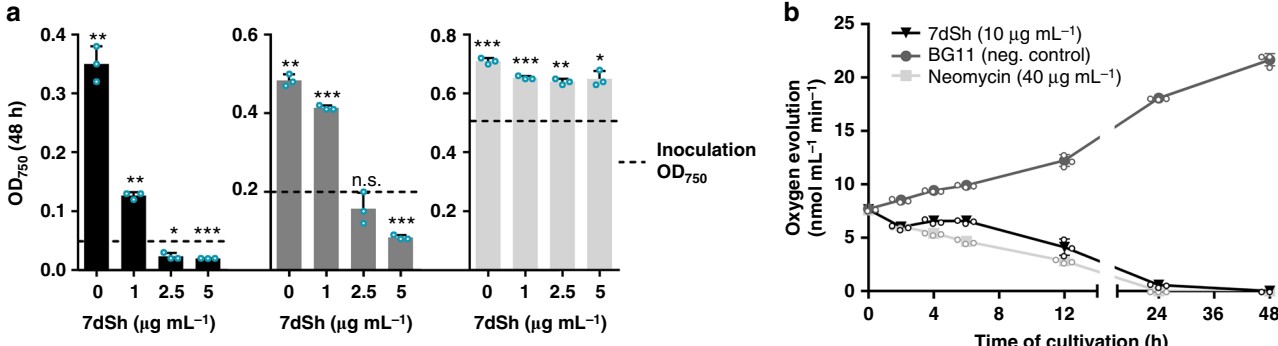

**Fig. 3** Effect of 7dSh (**1**) on the growth and photosynthetic oxygen evolution of *A. variabilis* cultures. **a** Growth of *A. variabilis* ($OD_{750}$) at different concentrations of 7dSh after 48 h of incubation. Cultures were inoculated to an $OD_{750}$ of 0.05, 0.2, or 0.5 (marked by dashed lines). 7dSh in aqueous solution was added at time 0. Significant differences between adjusted initial $OD_{750}$ and $OD_{750}$ after 48 h were analyzed in a one sample *t*-test (\**p*-value < 0.05; \*\**p*-value < 0.01; \*\*\**p*-value < 0.001; n.s., not significant). **b** Photosynthetic oxygen evolution by *A. variabilis* (initial $OD_{750} = 0.3$) in the presence of 7dSh or neomycin (positive control) or without supplementation (BG11, negative control). 7dSh (ca. 50 μM) and neomycin (ca. 65 μM) were added as aqueous solution. Values in both graphs represent the mean values of three biological replicates; standard deviations are indicated. Dots indicate data distribution. Source data are provided as a Source Data file

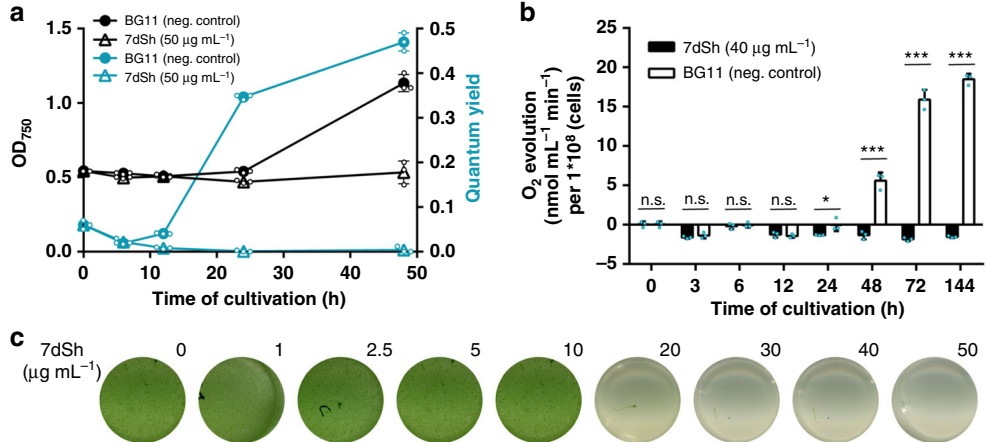

**Fig. 4** 7dSh (**1**) prevents regeneration of resuscitating *Synechocystis*. **a** Optical density (black) and PSII quantum yield (turquoise) of chlorotic *Synechocystis* cultures (initial $OD_{750} = 0.5$) regenerating in the absence or presence of 7dSh. $NaNO_3$ (17.3 mM) and 7dSh (ca. 260 µM) were added in aqueous solution at 0 h. Values represent the mean values of three biological replicates; standard deviations are indicated. Dots indicate data distribution. **b** Oxygen evolution of resuscitating *Synechocystis* cultures (initial $OD_{750} = 0.5$) upon addition of nitrate ($NaNO_3$, 17.3 mM) in the presence or absence of 7dSh (ca. 206 µM). Significant differences between 7dSh treatment and untreated control for each timepoint were analyzed in an unpaired *t*-test (\**p*-value < 0.05; \*\**p*-value < 0.01; \*\*\* *p*-value < 0.001; n.s., not significant). Values represent the mean values of three biological replicates; standard deviations are indicated. Dots indicate data distribution. **c** Cultures of chlorotic *Synechocystis* (initial $OD_{750} = 0.5$) 48 h after addition of nitrate ($NaNO_3$, 17.3 mM) and 7dSh. Numbers indicate concentration (µg mL$^{-1}$) of 7dSh added to the culture. Source data are provided as a Source Data file

neomycin, which slowly decreases photosynthetic oxygen evolution by inhibiting the PSII repair cycle. This similarity suggested an indirect effect of 7dSh on photosynthesis, ultimately mediated by the inability to maintain the PSII repair cycle.

To narrow down the cellular processes targeted by 7dSh (**1**), we made use of the unique properties of the recovery of nitrogen-starved chlorotic cells as an experimental system, where different metabolic activities are activated in a sequential order[34]. Here, long-term nitrogen-starved *Synechocystis* sp. cells were allowed to resuscitate from chlorosis by adding nitrate. In a typical experiment, the cells return to vegetative growth within 48 h in a highly coordinated process. Almost immediately after nitrate addition, dormant cells switch on metabolism and re-establish the basic enzymatic machinery. After approximately 16 h, photosynthesis and $CO_2$ fixation are turned on, and at the end of recovery, cells divide again. To reveal whether and at which stage resuscitation is blocked by 7dSh, chlorotic *Synechocystis* sp. cells were treated with 7dSh immediately before nitrate was added to initiate resuscitation. Following the addition of nitrate, control cultures showed the expected re-greening and return of photosynthetic activity[34]. The presence of 7dSh prevented resuscitation and re-greening in a dose-dependent manner (Fig. 4c). Measurement of oxygen exchange (Fig. 4b) and of PSII activity through pulse amplitude modulation (PAM) fluorometry[35] (Fig. 4a) showed that 7dSh-treated cells initially started respiratory glycogen consumption but then were unable to proceed further in the recovery and to restore their photosynthetic machinery. This clearly indicated that 7dSh affected metabolism at an early stage of resuscitation that is mainly characterized by anabolic reactions such as de novo amino acid synthesis[34].

**Inhibition of 3-dehydroquinate synthase by 7dSh**. To elucidate the mechanism of action, the effect of 7dSh (**1**) on the metabolic pattern of resuscitating *Synechocystis* sp. and exponentially growing *A. variabilis* was analyzed. Liquid cultures of the respective cyanobacteria were incubated in the absence or presence of 7dSh. At different time points, cells were collected and

extracted with an acidic methanol/water solution (see Methods) for molecular analysis by LC-HRMS. Software-based subtraction (MetaboliteDetect 2.1, Bruker Daltonics) of the standardized MS chromatograms facilitated the detection of metabolic differences between cell samples of untreated and 7dSh-treated cultures. This analysis revealed a fast and massive accumulation of a metabolite with the sum formula of $C_7H_{13}O_{10}P$ ($M_R = 288.14$ Da from $m/z = 289.0325$ $[M+H]^+$ and $287.0171$ $[M-H]^-$) in 7dSh-treated cells. Within 1 h after 7dSh (**1**) addition to *A. variabilis* cultures, the concentration of the respective compound increased more than fifteen fold as compared to initial concentration of untreated control cultures ($t = 0$ h) (Supplementary Fig. 3a, b). The accumulation of the respective compound further increased over time, reaching the 72-fold concentration (about 1.1 µM) as compared to untreated control cultures (about 16 nM) after 4 h. The sum formula ($C_7H_{13}O_{10}P$) and comparison of the MS/MS fragmentation pattern (Supplementary Fig. 3c) with MetFrag insilico fragmentation[36] and data in the literature[37] revealed that the accumulated compound was 3-deoxy-D-*arabino*-heptulosonate 7-phosphate (DAHP) (**4**, Supplementary Fig. 4). DAHP is the substrate of 3-dehydroquinate (DHQ) synthase, one of the first enzymes in the shikimate pathway, which converts DAHP to DHQ. This essential reaction in shikimate biosynthesis cannot be bypassed by alternative enzymes. The accumulation of DAHP is in accordance with DHQ synthase being the biological target of 7dSh[38]. Within the five-step reaction mechanism for conversion of DAHP to DHQ by DHQ synthase[39], the second step represents the β-elimination of the phosphate group of DAHP (Supplementary Fig. 4). We propose that 7dSh (**1**) mimics DAHP (**4**), the natural substrate of DHQ synthase. The C-7-methyl group of 7dSh, which is absent in DAHP, would impede the β-elimination in step 2, thereby leading to an inhibition of DHQ synthase and consequently the accumulation of DAHP.

Inhibition of the shikimate pathway triggers a metabolic perturbation that leads to decreased pools of aromatic amino acids, and, as a result of perturbed protein synthesis, also to the accumulation of non-aromatic amino acids such as leucine, valine, and arginine[40]. Thus, to confirm our hypothesis, we analyzed the levels of aromatic and selected non-aromatic amino

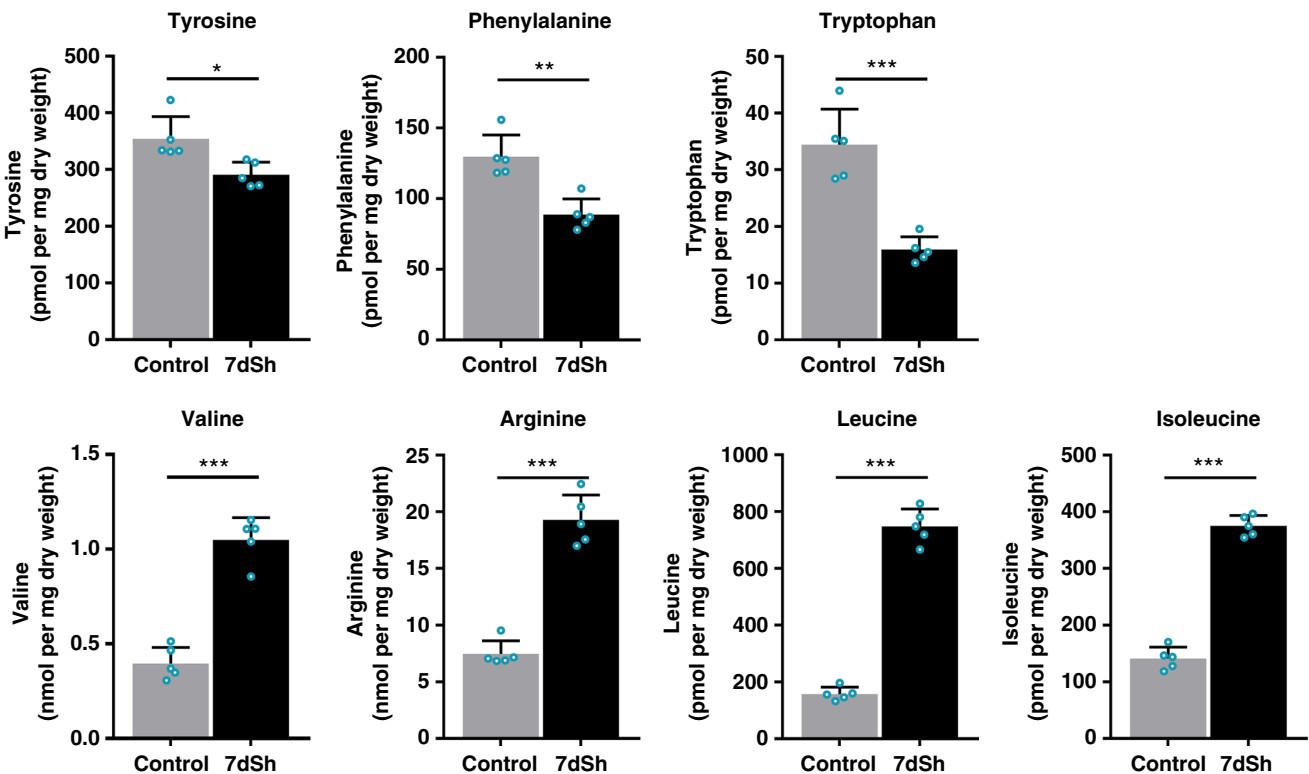

**Fig. 5** Effects of 7dSh (**1**) on amino acid levels in *A. variabilis* cells. Levels of selected amino acids in *A. variabilis* (initial $OD_{750} = 0.5$) treated with 7dSh (260 μM) for 4 h and respective untreated control cultures. Significant differences between 7dSh treatment and untreated control were analyzed in an unpaired *t*-test (\**p*-value < 0.05; \*\**p*-value < 0.01; \*\*\**p*-value < 0.001; n.s., not significant). Values represent the mean values of five biological replicates; standard deviations are indicated. Dots indicate data distribution. Source data are provided as a Source Data file

acids in 7dSh-treated and untreated *A. variabilis* cultures by LC-HRMS (Fig. 5). 7dSh (**1**) induced a significant accumulation of the non-aromatic amino acids leucine, isoleucine, valine and arginine. Within 4 h, the levels of isoleucine, arginine and valine increased almost threefold, and that of leucine about fivefold. By contrast, the levels of all aromatic amino acids significantly decreased in comparison to untreated control cultures (about 55% for tryptophan, 30% for phenylalanine and 20% for tyrosine).

The significant accumulation of DAHP (**4**) and changes in amino acid levels were also detected in cultures of nitrogen-starved *Synechocystis* sp. that were resuscitating from chlorosis in the presence of 7dSh (**1**) (Supplementary Fig. 5). Because of the lower metabolic activity of chlorotic cultures, DAHP accumulation was delayed but comparable to 7dSh-treated *A. variabilis* cultures.

We obtained further evidence that 7dSh (**1**) is an inhibitor of the shikimate pathway in an amino acid feeding experiment. The uptake of aromatic metabolites should mitigate the effects induced by 7dSh. PAM fluorometry of *A. variabilis* cultures revealed a 7dSh-induced decrease in the PSII quantum yield to about 10% of that of untreated control cultures (Supplementary Fig. 6). This effect was alleviated by supplementation with a mixture of aromatic amino acids. By contrast, supplementation of untreated control cultures with the aromatic amino acid mixture did not affect their PSII quantum yield. Supplementation with aromatic amino acids similarly alleviates the effects of glyphosate on other cyanobacteria[41].

**Antifungal and herbicidal effects of 7dSh**. As the shikimate pathway occurs in other bacteria and in fungi and plants, we

decided to investigate the effects of 7dSh (**1**) on organisms other than cyanobacteria. We chose the yeast model organism *Saccharomyces cerevisiae* as the fungal representative. When *S. cerevisiae* was cultivated in YPD complex medium, 7dSh did not affect growth. By contrast, when the yeast was grown in YNB minimal medium with defined carbon and nitrogen sources, 7dSh (10 μg mL$^{-1}$, ca. 50 μM) inhibited growth (Supplementary Fig. 7), with a lower growth rate and a significantly lowered final optical density ($OD_{600}$ of about 0.5) compared to growth in the absence of 7dSh (final $OD_{600}$ of about 0.95). Glyphosate (100 μg mL$^{-1}$, ca. 590 μM) had to be applied at more than tenfold higher concentration to achieve a similar effect. A similar decreased growth rate instead of complete growth inhibition of microbes has earlier been described for glyphosate[42]. If an anti-metabolite binds reversibly to the targeted enzyme, it will be replaced by the accumulating natural substrate. Therefore, a high intracellular concentration of the antimetabolite and low abundance of target enzyme favor inhibition of the targeted reaction. The observed residual growth of *S. cerevisiae* is consistent with a putative reversible binding of 7dSh.

As the representative for testing the effects of 7dSh (**1**) on plants, we chose the model organism *Arabidopsis thaliana*. Seedlings of *A. thaliana* germinated in mineral salt medium were significantly affected by micromolar concentrations of 7dSh. After 7 days, seedlings of the untreated control formed distinct roots with numerous root hairs and green cotyledons (Fig. 6a). The control seedlings showed gravitropism, and the distance between the shoot and root apical meristem was about 6 mm (Fig. 6b). Even a low concentration of 7dSh (5 μg mL$^{-1}$, ca. 25 μM) significantly affected the size of the seedlings. At concentrations of 25 or 50 μM, the growth inhibition effects of 7dSh were similar to those of glyphosate at the same concentrations.

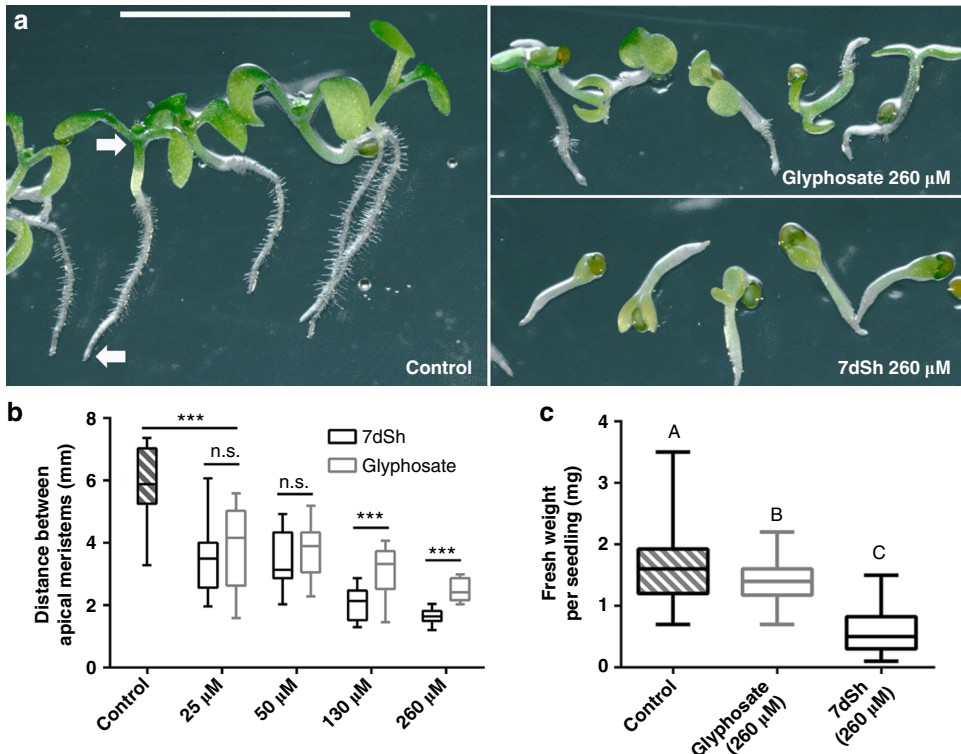

**Fig. 6** 7dSh (**1**) reduces the growth of *A. thaliana* seedlings. **a** Morphological appearance of autotrophically grown *Arabidopsis thaliana* seedlings 7 days after induction of germination. Seedlings were grown in constant light on agar plates without an antimetabolite (control) or in the presence of 7dSh or glyphosate. Plates were mounted vertically and illuminated from above. White arrows mark the root and shoot apical meristem. Scale bar (5 mm) applies to all images. **b** Measurement of the distance between root and shoot apical meristem. Significant differences between seedling sizes were analyzed in an unpaired *t*-test (*$p$-value < 0.05; ** $p$-value < 0.01; *** $p$-value < 0.001; n.s., not significant). Box-and-whisker plots represent the values of at least 16 seedlings. **c** Effect of 7dSh and glyphosate (each 260 µM) on the growth of *A. thaliana* on soil after 18 days in a day/night cycle. Statistical analysis was performed by using a one-way ANOVA. Tukey's multiple comparison test was used as the post-hoc test. Means that were significantly different ($p$-value < 0.05) are marked with different capital letters in the diagram. Box-and-whisker plots represent the values of at least 58 *A. thaliana* seedlings. For **b, c**: Error bars indicate range, box bounds indicate second and third quartiles, center lines indicate median. Source data are provided as a Source Data file

At concentrations of 130 and 260 µM, the inhibitory effect of 7dSh significantly surpassed that of glyphosate at the same concentrations, both in terms of seedling size (Fig. 6b) and morphological appearance (Fig. 6a). Impaired growth and aberrant morphology of the seedlings were particularly evident at higher concentrations of 7dSh (260 µM). In this case, the seedling growth was arrested within the first days. Only minor root and cotyledon formation was observed, and gravitropism was impaired (Fig. 6a and b). In comparison, *A. thaliana* seedlings that had been treated with higher concentration of glyphosate (260 µM) were less affected. They developed further, formed roots with small root hairs and bigger cotyledons. In the following days (day 7 to 14), no further plant growth or morphological change was observed in the presence of the inhibitors.

LC-HRMS analysis of whole plant extracts of the *A. thaliana* seedlings revealed a 7dSh-induced accumulation of DAHP, which was not detectable for the control or glyphosate-treated seedlings (Supplementary Fig. 8). As a proof of principle the accumulation of shikimate 3-phosphate was detectable in glyphosate-treated seedlings but not detectable in control or 7dSh-treated plants.

In order to evaluate the herbicidal activity of 7dSh (**1**) in more natural conditions, growth of *A. thaliana* in presence of the inhibitor was investigated in soil in a day/night cycle (Fig. 6c). After 18 days, seedlings were harvested and weighted. The weight

of the seedlings was significantly reduced in 7dSh and glyphosate treatment as compared to untreated control. Furthermore, the inhibitory effect of 7dSh significantly surpassed that of glyphosate as the weight of the 7dSh-treated seedlings was less than half as much as that of the glyphosate-treated seedlings.

Early germination events are characterized by the efficient reactivation of metabolic pathways[43]. Metabolites required for the induction of germination are stored in the seeds. Once these reserve materials are depleted, the proliferation of the seedlings relies on de novo synthesis of intermediates and growth factors. Glyphosate- or 7dSh-induced inhibition of the shikimate pathway therefore leads to an effective arrest of the seedling growth.

**Cytotoxicity of 7dSh on mammalian cells**. To determine whether 7dSh (**1**) affects mammalian metabolism, we tested various human cell lines (THP-1 macrophages, A549 human lung epithelial cells, HepG2 human liver epithelial-like cells, 293 human embryonal kidney cells) and primary human neutrophils in cytotoxicity assays. 7dSh did not show any cytotoxic effects on tested human cell lines and primary cells (Supplementary Fig. 9a), even at 5 mM, a concentration that is two orders of magnitude higher than that required for its herbicidal effect. Neither 7dSh nor glyphosate at 5 mM lysed cells, as measured by the release of lactate dehydrogenase. Further, the morphological appearance

of 7dSh-treated macrophages did not differ from that of the untreated control (Supplementary Fig. 9b,c).

**The biology of 7dSh for the producer strain**. It is surprising to us that 7dSh (**1**), which has never before been described in cyanobacteria, was isolated from the unicellular cyanobacterium *S. elongatus*. *S. elongatus* is a common laboratory strain but has never been described as a producer of hydrophilic secondary metabolites. The role of 7dSh in metabolism and for the physiology of the producer strain yet remains enigmatic. Due to the streamlined genome, which lacks classical secondary metabolite gene clusters[25], the biosynthesis of 7dSh is not yet clear. However, to the best of our knowledge, a specific biosynthetic gene cluster or pathway may not be necessary for the biosynthesis of 7dSh. It has been shown for another cyanobacterial strain with a small genome, that enzymatic promiscuity enables the production of a large variety of secondary metabolites without the need of specific enzymes[23]. One of the enzymes known for enzymatic promiscuity is the transketolase, the enzyme we used for chemoenzymatic synthesis of 7dSh. The transketolase plays a fundamental role in cyanobacterial metabolism, e.g., in the Calvin cycle, and exhibits a wide substrate specificity. To shed light on the biosynthesis of 7dSh, we screened *S. elongatus* cultures for the presence of a potential 7dSh precursor. The screening resulted in the isolation of the monosaccharide 5-deoxy-D-ribose (**2**), a metabolite never isolated from nature before (Supplementary Fig. 10a, b, Supplementary Fig. 17–21).

The effective in vitro conversion of 5-deoxy-D-ribose (**2**) to 7dSh (**1**), carried out by the *S. elongatus* transketolase in our chemoenzymatic synthesis, suggests this reaction is the final step in the biosynthesis of 7dSh. The affinity of the cyanobacterial transketolase to 5-deoxy-D-ribose was about 100-fold lower than the affinity to the natural transketolase substrate D-ribose 5-phosphate (Supplementary Fig. 10c, d), enough to explain the minute levels of 7dSh produced by *S. elongatus* in later stages of growth, where $CO_2$ fixation and consequently D-ribose 5-phosphate levels decrease. Nevertheless, the synthesis of the assumed precursor 5-deoxy-D-ribose remains enigmatic. Future studies shall clarify whether this compound is a side product of a primary metabolic pathway or whether yet unidentified enzymes are involved.

The biological role of 7dSh (**1**) for the producer strain remains obscure. Since 7dSh shows allelochemical characteristics and is excreted to the medium, its production could also be a strategy of *S. elongatus* to protect its niches against other competitors. Even though the low production level of 7dSh in laboratory test conditions questions this hypothesis, the 7dSh concentration could be higher under certain natural conditions, such as in biofilms. Although *S. elongatus* laboratory strains usually grow planktonically, the re-isolated wild-type grows in a biofilm[44]. When growing in a biofilm or a microbial mat, the concentration of 7dSh could increase to a level sufficient to provide physiological effects such as controlling the surrounding microbial community.

## Discussion

This paper reports the first natural compound—the rare sugar 7-deoxy-sedoheptulose (7dSh; **1**)—that acts as an in vivo inhibitor of the DHQ synthase as part of the shikimate pathway. Even though various inhibitors of DHQ synthase have been described in the literature[38,45–47] (Supplementary Fig. 11), none of these metabolites were reported to show strong in vivo activity against microbes or plants. Furthermore, in contrast to the known DHQ synthase inhibitors, 7dSh is comparatively simple to synthesize and production could be easily scaled up. Notably, 7dSh shows

in vivo activity, especially against autotrophically growing organisms. We expect that 7dSh would also inhibit the growth of microorganisms in habitats with minimal nutrient concentrations. In nutrient-saturated habitats, microorganisms can take up aromatic metabolites from their environment and thereby mitigate the effect of 7dSh. The promising in vivo activity confers 7dSh antibacterial, antifungal, and herbicidal characteristics, which could enable its deployment as an agent in agriculture, water management, veterinary medicine, and even human medicine. Additional studies are needed to determine the medical and economic potential of 7dSh.

## Methods

**Strains and culture conditions**. Cyanobacterial strains (*Synechococcus elongatus* PCC 7942, *Anabaena variabilis* ATCC 29413, and *Synechocystis* sp. PCC 6803) were routinely cultivated under photoautotrophic conditions with continuous illumination at 30–60 µE (Lumilux de Lux, Daylight, Osram) at 27 °C. Cells were cultivated in flasks with shaking at 120–130 rpm. Unless indicated otherwise, cells were cultivated in BG11 medium[48] supplemented with 5 mM NaHCO3. For BG11 solid medium, 15 g L$^{-1}$ Bacto agar (Difco) was autoclaved separately and combined with BG11 liquid medium.

Large-scale batch cultures of *S. elongatus* were cultivated in 1 L flasks containing 700 mL BG11 medium for 14 days under illumination at 55 µE. The cultures were gassed with air supplemented with 2% $CO_2$. Batch cultures were inoculated with densely grown pre-cultures to an OD$_{750}$ of 0.2.

For cultivation of nitrogen-starved *Synechocystis* sp., NaNO3 was omitted from BG11 medium as previously described[34]: Nitrogen starvation was initiated by centrifugation of the cells at room temperature at less than $3500 \times g$ (Hereaus Megafuge 1.0 R). Cell pellets were washed, resuspended (desired OD$_{750}$ = 0.4) and cultivated in BG11 without NaNO3 for two weeks (shaken in 500 mL Erlenmeyer flasks). Resuscitation of chlorotic cells was initialized by the centrifugation of the cells at less than 3500 xg (Hereaus Megafuge 1.0 R) followed by the resuspension of the cell pellets in BG11.

*Streptomyces setonensis* SF666 (NBRC No. 13797) and *Gluconobacter oxydans* subsp. *suboxydans* (VTT E-97003) were cultivated as previously described[33]. Briefly, *S. setonensis* was grown in complex media containing 2.5% (w/v) glucose, 3.5% (w/v) soy flour (soybean meal), 0.5% (w/v) soluble vegetable protein and 0.25% (w/v) NaCl at pH 7.0 and 28 °C for 7 days. Cultures were grown under constant shaking (250 rpm) in 500 mL Erlenmeyer flasks covered by foam caps. *G. oxydans* was grown in complex media containing 0.2% (w/v) Na-glutamate, 0.2% (w/v) K2HPO4, 2% (w/v) sucrose, 0.2% (w/v) peptone, 0.5% (w/v) yeast extract, 0.01% (w/v) MgCl2, 0.001% (w/v) FeSO4 and 0.001% (w/v) MnSO4 at pH 6.8 and 30 °C. Cultures were tested for their sensitivity against 7dSh on agar plates (1.5% (w/v) agar) and in liquid media in 96 well plates. 96 well plates were analyzed by a microplate reader (Tecan Spark®10 M) at a wavelength of 600 nm.

*Saccharomyces cerevisiae* was grown in yeast extract-peptone-dextrose (YPD)[49] medium or yeast nitrogen base (YNB) without amino acids (Sigma-Aldrich) medium supplemented with 0.5 g L$^{-1}$ fructose and 1 g L$^{-1}$ casamino acids with continuous shaking at 30 °C.

For agar plate experiments seeds of *Arabidopsis thaliana* accession Col-0 were germinated in half-strength Murashige and Skoog (MS) salts basal medium (Sigma Aldrich) agar plates (1.5%, w/v, Bacto agar) under constant illumination (60 µE) at 24 °C. For simultaneous growth of seedlings, seeds were stored at 4 °C overnight prior to initiation of germination. Subsequent, seedlings were grown for 7 days. To achieve growth of seedlings along the agar, plates were mounted vertically and illuminated from above.

For *Arabidopsis thaliana* experiments in soil, 24-well plates (three plates for each condition) were filled half with autoclaved and dried GS90 standard soil (Patzer GmbH, Germany) and vermiculite. Subsequently, wells were filled with 750 µL water (control), 7dSh or glyphosate (each 260 µM in aqueous solution, for glyphosate pH 7 was adjusted with NaOH). Each well was planted with a single seed of *A. thaliana* and the plate was incubated at 4 °C in the dark for five days to ensure a simultaneous growth of the seedlings. After that the seedlings were transferred to a growth chamber (air humidity 40%) with a 16 h day (20 °C) and 8 h night (18 °C) cycle with a light intensity of 85 µE. After 18 days the seedlings were harvested and weighted.

**Extraction of *S. elongatus* culture lyophilisates**. For lyophilisate extraction, 100 mL of *S. elongatus* batch cultures (grown for 14 days) were lyophilized. Lyophilisate was solved in 1 mL methanol, chloroform, acetone, or ethyl acetate. A 10 µL aliquot of each extract was applied to agar diffusion plates spread with *A. variabilis*.

**Correlation of OD$_{750}$ and inhibitor production level**. At each time point, 25 mL of each *S. elongatus* batch culture was centrifuged at room temperature at $4500 \times g$ for 5 min. The supernatant was evaporated to dryness, and the residue was dissolved in 80 µL methanol. A 40 µL aliquot of each extract was applied to agar diffusion plates spread with *A. variabilis*.

**A. variabilis growth inhibition assays**. The inhibition of the growth of *A. variabilis* by *S. elongatus* extracts was assayed in BG11 liquid medium and on BG11 solid medium. In agar diffusion tests, paper discs containing dried samples of *S. elongatus* extracts were applied to agar plates freshly inoculated with *A. variabilis*. Agar plates were incubated under constant illumination at 40 µE and 27 °C for 5–6 days.

For growth inhibition assays in liquid medium, *A. variabilis* was grown in 24-well plates in 1 mL BG11 (initial OD = 0.05, unless stated otherwise). Test samples and controls were applied in water. The cultures were shaken at 100 rpm and 27 °C for 2–3 days under constant illumination at 40 µE.

**Summary of 7dSh purification from S. elongatus**. Culture supernatants were adjusted to pH 4 with 0.5 M HCl and then lyophilized. The lyophilisate was extracted with methanol and concentrated in vacuo. The methanol extract was applied to a gel filtration/size-exclusion column (Sephadex LH20, 1.6 × 80 cm, flow rate 0.5 mL min$^{-1}$, in methanol). The active fractions ($t_R$ ca. 8 h) were pooled, evaporated to dryness and loaded in silica gel onto a Si 35, SF10–4g cartridge for separation of metabolites by normal phase medium-pressure liquid chromatography (MPLC) at a flow rate of 10 mL min$^{-1}$ with a chloroform (solvent A) /methanol (solvent B) gradient as follows: 100% A, then solvent B in solvent A increased by 10% every 5 min to a total of 40% B in 25 min. The active fractions (elution after about $t_R$ = 18–21 min) were pooled, evaporated to dryness, re-dissolved in water and loaded onto a ligand/ion-exchange high-performance liquid chromatography (HPLC) column (HiPlex Ca, 300 mm × 7.7 mm, Agilent). HPLC with isocratic water elution (flow 0.5 mL min$^{-1}$, temperature column oven: 85 °C) for 20 min led to the chromatographically pure 7dSh (**1**) ($t_R$ = 15 min).

**Physiochemical characterization of 7dSh from S. elongatus**. For high-resolution mass spectrometry (HR-MS) data, purified 7dSh (**1**) was applied to a HiPlex Ca column of a Dionex Ultimate 3000 HPLC system (Thermo Fisher Scientific) coupled to a maXis 4 G ESI-QTOF mass spectrometer (Bruker Daltonics). 7dSh was isocratically eluted with water (flow rate 0.5 mL min$^{-1}$, temperature column oven: 85 °C) for 20 min. The ESI source was operated at a nebulizer pressure of 2.0 bar, and dry gas was set to 8.0 L min$^{-1}$ at 200 °C. MS/MS spectra were recorded in auto MS/MS mode with collision energy stepping enabled. The scan rates for full scan and MS/MS spectra were set to 1 Hz and 7 Hz, respectively. Sodium formate was used as internal calibrant in each analysis. The molecular formula was calculated from monoisotopic masses using the SmartFormula function of DataAnalysis (Bruker Daltonics).

NMR measurements were recorded on a Bruker AMX600-, Avance III HDX700-, and AVI400 instruments. Deuterated methanol or water was used as solvent and internal standard. All spectra were recorded at room temperature. Chemical shifts are reported as δ values relative to the respective solvent as an internal reference. Coupling constants (*J*) were reported in Hertz (Hz). Abbreviations of multiplicity: *s* = singlet, *d* = doublet, *dd* = doublet of doublet, *m* = multiplet. Optical rotations were measured with a Perkin–Elmer 241. *Rf* values on TLC were determined on silica gel 60 F254 plates (Merck, 0.2 mm). Compounds were detected with orcinol staining reagent (10 mL sulfuric acid containing 0.1 g Fe (III)-chloride and 1 mL orcinol solution (in 6% ethanol)).

**Cloning and purification of S. elongatus transketolase**. The ORF *Synpcc7942_0538* of *Synechococcus elongatus* PCC 7942 was PCR amplified using the primer combination 5′-CATCACAGCAGCGGCCTGGTGCCGCGCGGCAG CCATATGCTCG AGATGGTTGTTGCGGCTCAATC-3′ and 5′-AGCAGCCA ACTCAGCTTCCTTTCG GGCTTTGTTA GCAGCCGGATCCTAGCCGATCA CTGCTTTCG-3′. After restriction of the pET15b vector with *Bam*HI, the fragment was fused with the vector backbone using Gibson assembly[50]. The final construct was introduced into *E. coli* BL21 (DE3) cells by transformation. Overexpression of *Synpcc7942_0538* was induced by addition of 0.5 mM IPTG. After 12 h of induction at 37 °C, cells were harvested by centrifugation. All subsequent steps were carried out at 0–4 °C. The pellet was resuspended in 10 mL of lysis buffer (50 mM Tris-HCl, 300 mM NaCl, 10 mM imidazole, pH 7.5, protease inhibitor (complete ULTRA tablets, Roche), lysozyme, and DNAse), and cells were lysed by sonication. Cell debris and insoluble material were removed by centrifugation (25 min at 4 °C and 35,000 × *g*). The cleared cell lysate was applied to a HisTrap HP column (GE Healthcare Life Science). After column washing, the bound proteins were eluted in fractions with 10 mL elution buffer (50 mM Tris-HCl, 300 mM NaCl, 500 mM imidazole, pH 7.5). Transketolase-containing fractions were combined, dialyzed in dialysis buffer (50 mM Tris-HCl pH 8.0, 100 mM NaCl, 5 mM MgCl₂, 1 mM DTT, 0.5 mM ETDA, 50% glycerol) and stored at −20 °C.

**Chemoenzymatic synthesis of 7-deoxy-ᴅ-altro-2-heptulose**. The synthesis of 7dSh (**1**) was inspired by the chemoenzymatic synthesis of sedoheptulose 7-phosphate[51]. 5-Deoxy-ᴅ-ribose (**2**) (Glentham Life Sciences) (50 mg, 250 mM) was dissolved in 1.5 mL HEPES buffer (100 mM, pH 7.5) containing thiamine pyrophosphate (1.3 mg, 2 mM) and MgCl₂ (0.4 mg, 3 mM). β-Hydroxypyruvate (**3**) as its lithium salt hydrate (54 mg, 285 mM) was added, and the pH was adjusted to 7.5. The reaction was initiated by addition of 4 mg transketolase (EC 2.2.1.1), and the mixture was shaken at 400 rpm and 30 °C for 24 h (Thriller®, Peqlab). The

reaction was stopped by the addition of 6 mL methanol, followed by centrifugation (2500 × *g*, 10 min). The supernatant was evaporated to dryness, and 7dSh was purified as described in our purification protocol for extraction of 7dSh from *S. elongatus* cultures, except that size-exclusion chromatography on Sephadex LH20 was omitted.

**Photosynthetic oxygen evolution and PAM fluorometry**. Photosynthetic oxygen evolution was determined in vivo using a Clark-type oxygen electrode (Hansatech Instruments RS232, Norfolk, UK). Two milliliters of treated or untreated cultures were transferred to the measurement chamber. Oxygen formation was measured for 10 min at room temperature under illumination at 50 µE. PSII activity was analyzed in vivo with a WATER- pulse amplitude modulation (PAM) chlorophyll fluorometer (Walz GmbH, Effeltrich, Germany). All samples were dark-adapted for 5 min before measurement. Measurement of the PSII quantum yield ($F_v/F_m$) was performed at room temperature with WinControl Data Acquisition software.

**Analysis of cyanobacterial and herbal metabolite patterns**. Aliquots (2 mL) of cyanobacterial cultures were centrifuged (30 s, 20,817 × *g*), and the pellets were immediately frozen in liquid nitrogen. *A. thaliana* plants were crushed in a mortar under liquid nitrogen cooling. The pellets or plant materials were extracted with 600 µL methanol, followed by a second extraction of the material with 600 µL 20% methanol containing 0.1% formic acid. Both supernatants of each material extraction were combined and lyophilized, and the residue was dissolved in 100 µL 20% methanol containing 0.1% formic acid. Extracts (10 µL) were examined by LC-HRMS (Dionex Ultimate 3000 HPLC system from Thermo Fisher Scientific, coupled to a maXis 4 G ESI-QTOF mass spectrometer from Bruker Daltonics). LC-HRMS settings were adopted from[52]. Obtained mass spectra were compared and processed using MetaboliteDetect (Bruker Daltonics). Molecular formulas were calculated from monoisotopic masses using the SmartFormula function of DataAnalysis (Bruker Daltonics).

**Quantification of amino acids and DAHP in A. variabilis**. For the quantification of amino acids (Trp, Tyr, Phe, Val, Arg, Leu, and Ile) and DAHP the freeze dried sample material (10 mL, OD$_{750}$ = 0.5) was homogenized with a Retsch ball mill (two cycles, 30 s each). Extraction was done with 400 µl 80% methanol containing 0.1% formic acid followed by a second extraction step with 400 µl 20% methanol also containing 0.1% formic acid. Both supernatants were combined and brought to dryness in a vacuum concentrator. The dried samples were redissolved in 150 µl 0.1 M hydrochloric acid for analyses.

Amino acid analyses were done with a Water UPLC-SynaptG2 LC-MS system. Chromatography was carried out on a 2.1 × 100 mm Waters Acquity HSST3 column. For separation a 10 min gradient from 99% water to 99% methanol (both solvents with 0.1% formic acid) was used. The mass spectrometer was operated in ESI positive mode and scanned from 50 to 2000 *m/z* with a scan rate of 0.5 s. For quantification extracted ion chromatograms were generated and integrated. An external calibration function was used for the calculation of absolute amounts.

DAHP analyses were done on a Thermo Scientific/Dionex ICS 5000 system. Chromatography was carried out on a 3.0 × 150 mm Carbopac PA 20 column. For separation a 32 min gradient from 75 mM sodium hydroxide to a 75 mM sodium hydroxide/500 mM sodium acetat mixture was used. Quantification was done by integration of the signal and absolute amounts were calculated with an external calibration function.

**Supplementation with aromatic amino acids**. *A. variabilis* in BG11 liquid medium (initial OD$_{750}$ = 0.2) was supplemented with aromatic amino acids to a final concentration of 1 mM tryptophan, 1 mM tyrosine, 1 mM phenylalanine, 1 µg mL$^{-1}$ *p*-aminobenzoate, or 1 µg mL$^{-1}$ *p*-hydroxybenzoate.

**Cytotoxicity of 7dSh on mammalian cells**. Human THP1 cells (DSMZ, ACC 16) were grown in RPMI 1640 medium with 2 mM glutamine, 10% heat-inactivated FBS, 2% HEPES, 1% penicillin-streptomycin (10000 U mL$^{-1}$, Gibco) and 1 mM sodium pyruvate. To induce differentiation, cells were seeded in RPMI medium containing 1% penicillin-streptomycin and treated with 160 nM phorbol-12-myr-istate-13-acetate for 24 h. A546 cells, HepG2 cells, and HEK 293 cells were grown in Dulbecco's Modified Eagle's Medium (DMEM) with 10% Fetal Bovine Serum (FCS) and 1% penicillin-streptomycin. Human neutrophils were isolated from healthy blood donors by biocoll/histopaque density gradient centrifugation and resuspended in RPMI + 2% human serum albumin (HSA) + 2 mM sodiumpyr-uvat + 10 mM HEPES. THP1 cells (1 × 10$^5$ or 3 × 10$^5$) were seeded in 96-well cell culture plates in a final volume of 100 µL or in 8-well µ-slides in a final volume of 300 µL, respectively. After differentiation, the cells adhered to the culture dishes. A546 (ATCC CCL-185) cells, HepG2 (ATCC HB-8065) cells, HEK 293 (InvivoGen 293-null) cells (0.5 × 10$^5$) and human neutrophils (1 × 10$^6$) were seeded in 96-well cell culture plates in a final volume of 200 µL.

To evaluate the cytotoxic potential of 7dSh (**1**) and glyphosate, human THP1 macrophages were treated with 5 mM of the respective compounds in RPMI medium with 1% penicillin-streptomycin for 24 h. A546 cells, HepG2 cells, and HEK 293 cells were treated with 5 mM of the respective compounds for 24 h in DMEM medium (without phenol red) + 10% FCS + 1% penicillin-streptomycin.

Human neutrophils were treated in RPMI + 2% HSA + 2 mM sodium pyruvate + 10 mM HEPES for 5 h. Cells in medium were used as negative control, and cells treated with 1% Triton X-100 were used as positive control (100% cytotoxicity). After the indicated time points the cytotoxic potential of the compounds was determined according to the release of lactate dehydrogenase into the supernatant using the Cytotoxicity Detection Kit (Roche) following instructions of the manufacturer. The data represent the mean of three independent experiments.

For microscopic analysis, THP1 macrophages in 8-well μ-slides were treated with 5 mM 7dSh in RPMI medium with 1% penicillin-streptomycin or were untreated for 24 h. Cells were fixed with 150 μl icecold PBS containing 3.7% formaldehyde for 30 min. Wells were washed three times with HBSS, and incubated with 2.5 μl Alexa Fluor 647 Phalloidin (Thermo Fisher) in 100 μl PBS containing 1% BSA for 30 min. Cells were stained with one drop of NucBlue® Fixed Cell ReadyProbes® reagent (DAPI, Thermo Fisher) and mounted using fluorescence mounting medium (DAKO). Image acquisition was performed in the confocal mode of an inverted Zeiss LSM 710 NLO microscope employing Zeiss Plan-Apochromat ×63/1.40 oil DIC M27 objective with the following excitation wavelengths: DAPI: 405 nm; Phalloidin: 633 nm. Images were exported in overlays as 16-bit tagged image files for further analysis. Overlays were batch processed for intensity and color balance.

**Isolation and structural elucidation of 5-deoxy-D-ribose**. 5-Deoxy-D-ribose (2) was isolated from cultures of *S. elongatus* as described for the isolation of 7dSh (1) from *S. elongatus* except that pooling of the fractions from chromatography was adapted to 5-deoxy-D-ribose. On Sephadex LH20, 5-deoxy-D-ribose co-eluted with 7dSh. On MPLC, 5-deoxy-D-ribose eluted after 14–15 min. On HPLC with a HiPlex Ca column, the compound showed a retention time of 25.5 min. The isolated compound and commercially available 5-deoxy-D-ribose (Glentham Life Sciences) were analyzed by NMR spectroscopy and mass spectrometry as described for 7dSh.

**Kinetic characterization of the *S. elongatus* transketolase**. The kinetics of the *S. elongatus* transketolase were characterized for D-ribose 5-phosphate (Sigma Aldrich) or 5-deoxy-D-ribose (2) (Glentham Life Sciences) as acceptors of the transferred C1-C2 ketol unit. Reaction mix contained 0.1 mM glycylglycine buffer pH 7.5, 3 mM MgCl$_2$, 2 mM thiamine pyrophosphate (TPP), 0.5 mM NADH, 100 mM-L-erythrulose, 10 units yeast ADH and 5.75 μg mL$^{-1}$ *S. elongatus* transketolase. For determination of $K_M$ and $V_{max}$, the substrate concentrations were varied: D-ribose 5 phosphate (0.1 mM to 2.0 mM) or 5-deoxy-D-ribose (75 mM to 600 mM). Absorption at 340 nm was recorded. The reaction was performed in a final volume of 200 μL or 400 μL. Michaelis–Menten kinetic profiles were determined using GraphPad prism.

**Reporting summary**. Further information on experimental design is available in the Nature Research Reporting Summary linked to this article.

## Data availability
Data supporting the findings of this work are available within the paper and its Supplementary Information files and from the corresponding authors upon reasonable request. The source data underlying Figs. 1, 3–6 and Supplementary Figs 3, 5, 6, 7, 9, and 10 are provided in Source Data file.

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

## Acknowledgements

Work of the authors is supported and funded by the RTG 1708 "Molecular principles of bacterial survival strategies", the Institutional Strategy of the University of Tübingen (Deutsche Forschungsgemeinschaft, ZUK 63) and the "Glycobiotechnology" initiative (Ministry for Science, Research and Arts Baden-Württemberg). We thank Dr. Klaus Eichele, Anorganic Chemistry, University of Tübingen for NMR-predictions. Thanks to Dr. Sandra Richter, ZMBP University of Tübingen, Dr. Dorothee Wistuba, University of Tübingen and Dr. Timo Niedermeyer, University of Halle and Dr. Christiane Wolz, IMIT Tübingen for helpful discussions. We are indebted to Karen Brune for critically reading the manuscript.

## Author contributions

K.B. performed culture cultivation, isolated 7dSh (**1**) and 5-deoxy-D-ribose (**2**), designed and performed synthesis of 7dSh and HPLC-MS experiments of the metabolite pattern and the biological test systems, wrote manuscript with input from all authors. J.R. designed and performed plant assays on soil and performed statistical analysis. P.R. performed NMR analysis and interpreted NMR data. A.S. designed and performed cloning, purification and kinetic characterization of transketolase. L.B. and E.W. designed and performed cytotoxicity studies on mammalian cells. M.S. designed and performed DAHP and amino acid quantification. S.G. designed and interpreted chemical experiments, edited manuscript. K.F. designed and interpreted biological experiments, edited manuscript.

## Additional information

**Competing interests:** University of Tübingen has filed a patent application that covers 7dSh, 7dSh analogs and their use (EKUT-0365, German patent application number DE10 2017 01 898.1, International patent application number PCT/EP2018/082440). The remaining authors declare no competing interests.

