## [Peer Review File · Nature Communications]

Reviewers' comments:

Reviewer #1 (Remarks to the Author):

The authors describe the discovery that 7-deoxy-sedoheptulose (7dSh) is an antimicrobial and herbicidal compound that apparently acts through inhibition of an early step of the shikimate pathway, 3-dehydroquinate synthase (DHQS). The work is well done and generally convincing.

Suggestions and comments

1. Comparison of the activity of 7dSh with that of a known DHQS inhibitor like carbaphosphonate would have been interesting.
2. I would prefer a briefer title. "Sweet natural killing" does grab the eye, but is 7dSh even sweet? 7dSh is not new, as it was first reported in 1970. The biological activity that is described is new. "Powerfully" is in the eye of the beholder. Compared to most commercial herbicides, it is not powerful. Glyphosate is not powerful either, as it is a high use rate herbicide compared to many others.
3. Lines 196-198. The phenomenon could also or partly be due to metabolic degradation or alteration of 7dSh by the cells.
4. Lines 225-227 – I disagree with this analysis. Similar results might be expected with inhibitors of targets in the chloroplast such as phytoene desaturase, HPPD, etc.
5. Fig. 6c is not referenced or discussed in the text. Please change the Y axis of Fig. 6c to molar measurements instead of peak areas. The amount of shikimate accumulating in glyphosate-treated plants is very large, leading some to think that inhibition of EPSPS leads to deregulation of the shikimate pathway, redirecting carbon from carbon fixation processes. So, do you see similar amounts of carbon accumulating as DAHP as you do as shikimate with glyphosate?
6. Petri dish assays of herbicides are often very misleading. Herbicides are used in two main ways – foliar sprays or in soil. Is there significant herbicidal activity when sprayed on plants? If not, is there any activity in seedlings germinating in soil?
7. I am not a toxicologist, but was there any effect on THP1 cell growth? Lines 385-387 – The contradictory results are mainly due to a retracted paper. The toxicologists who I know do not give credence to the reports of carcinogenic properties of glyphosate. See the special issue of Critical Reviews in Toxicology (Supplement 1 of volume 46 in 2016) for a full analysis. Provide scale bars in Fig. 7b. These micrographs are not very clear. I suggest relegation of Fig. 7 to supporting data.
8. Lines 377-379 – How can this statement be made unless these compounds are compared directly in the same bioassay with 7dSh and glyphosate?
9. Line 389 – The reference does not support the statement. There are papers that show that the half life of AMPA is longer than that of glyphosate in soil. But compared to many other pesticides, the half life of neither is a significant concern, although some have hyped the half life of AMPA.
10. Lines 391-392 – I am not clear as to why the taste of 7dSh supports the view that it has no undesirable side effects. Is it sweet? The title of the paper suggest that it is.

Wording suggestions and minor comments:

Line 63 - ..animals, especially mammals. I make this suggestion because humans are animals, as are insects, etc.

Line 70 - ..in arrested growth or...

Line 76 - ..initial commercialization in... It may have been approved earlier than 1974.

Lines 88 & 86 – The target is not very potent, but the antimetabolites are. So, change to:

..Another target of very potent antimetabolites....

Line 121 - ..content apparently peaked..

Line 292 – molar concentration?

Line 294 - ..inhibition of microbes is.. This is not true of glyphosate as a herbicide.

Line 314 – Change obstructed to impaired.

Line 317 – But, was growth inhibited during this time?

Line 324 – To many plant scientists, the seed has germinated when the radicle penetrates the seed coat. After that, what we see is seedling growth. So, I would say:germination and seedling growth process.

Line 379 – activity for what?

Line 387 – ‘Inherited’ is a biological term. Glyphosate has not inherited anything.

Line 392 – Delete pure

Line 646 –Ref. 14 - Do the authors have names?

Reviewer #2 (Remarks to the Author):

NCOMMS-18-20996-T Reviewer’s Coments

This is a generally well-written and comprehensive manuscript. I believe that it can become publishable after the following comments are addressed. Statistical analysis is particularly weak and must be remedied.

There are more than a few confusing sentences, mostly due to incorrect English usage and grammatical constructions. For example, line 62 ‘in their application’ is awkward, line 67 ‘in these organisms’ is unclear; line 77 ‘became’ should be ‘has become’; line 89 ‘this enzyme’ reference is unclear; 103 ‘collapsing’ ?; 141 a compound cannot prefer anything; 224 ‘reinstall’?; 314 ‘obstructed?’; etc etc.

The first paragraph of the introduction is inappropriate since there is no evidence that 7dSh plays any role in microbial communities or ecosystems, only speculation from the authors.

There is evidence that accumulation of shikimate 3-phosphate alone is inhibitory after glyphosate treatment.

Line

83: this sentence ignores all other inhibitors of the shikimate pathway, some of which are later referenced by the authors.

91: allelochemicals can be inhibitory to many other organisms besides just ‘closely related organisms’

110: this statement ignores the previously described SF-666B compound.

197: this argument implies irreversible binding, for which there is no evidence. In fact, the yeast results support a reversible binding mechanism. The authors also do not address the possibility of 7dSh catabolism, which is an equally likely explanation.

206: The slow decrease in photosynthesis could be compared to any number of inhibitors, not just neomycin.

272: Data have not been analyzed appropriately to make the claim that the delay was ‘not less significant’.

282: The referenced work only used glyphosate, not ‘other inhibitors of the shikimate pathway’.

311, 313: There are no data presented on seed germination, only seedling growth.

315: same as what?

316: ‘more distinct’ is unclear and not quantifiable

317: 7 days following what? Which inhibitors? Does this mean they all stopped growing or they reverted to untreated growth rates?

321: de novo was not italicized and was hyphenated on line 226

323: germination was completed in the presence of both inhibitors, but seedling growth was inhibited.

327: THP-1 cells are used mainly as in vitro cancer cell models and monocyte-macrophage differentiation processes, and thus are not the best model to study mammalian toxicity. Detecting cell lysis or observing morphological appearance are neither sensitive nor appropriate measures of toxicity. CHO or other cell lines would be a better choice. 7dSH effects on this one cell line cannot be used to extrapolate to other cell types or mammals.

338: considered by whom?
342: this sentence seems entirely speculative. Are there any data?
361: conceivable but entirely too speculative for this paper
362-369: this is nothing but speculation
378: sufficient for what?
380: up-scaled is not a word
386: are there references for these recent studies? The possible carcinogenicity of glyphosate is not 'intensely discussed' among knowledgeable herbicide physiologists, since reports of such activity are unreliable, unpublished, or have been retracted from refereed journals. What are 'inherited properties?'
389: 'High specificity and potent growth inhibition' are in no way related to a lack of undesirable side effects.
391: Cyanide smells like almonds—does that similarly support the expectation of a lack of undesirable side effects?
392: The allelopathy literature is replete with examples of 'pure natural products' that are extremely toxic and bioaccumulate.
393-397: The potential to develop this compound as a herbicide is remote. As done here, the addition of 7dSH to liquid growth medium bypasses all plant uptake and translocation requirements, which are not trivial. If this compound is being considered as a soil-applied herbicide (for root uptake), there are a host of complicating factors (formulation, degradation by soil microorganisms, persistence in soil, etc) that would have to be overcome. Foliar applications likewise present even more obstacles. It is enough for the authors to say that it is a potent inhibitor of the shikimate pathway, but extrapolation to commercial herbicide development is beyond the scope and should be deleted.

Figures 3 – 6, 7c: none of these data are analyzed correctly. SD bars by themselves cannot be used to make comparisons among treatments. Mean separations and statistical comparisons must be done.

Almost all figures have confusing or awkward axis labels.

Figure 5: 'Area of MS peak base' is confusing and inappropriate. These values should be compared to standard curves of known standards and expressed in biologically relevant units like umoles per mg of tissue. If the focus is the rate of accumulation or decline, data should be regressed over time with R values, rate constants calculated, and statistically compared.

Reviewer #3 (Remarks to the Author):

The authors present the results of a highly interesting study on a metabolite produced by *Synechococcus elongatus* that inhibits the growth of other cyanobacteria as well as of plants. The authors then isolate this substance and define it as an unusual deoxy-sugar. As such, it is an antimetabolite that interferes with shikimate biosynthesis. By measuring the build-up of unused substrate in inhibitor treated cultures, they deduced the site of inhibition to be 3-dehydroquinate synthase. Based on the structure of the inhibitor and its site of action, it appears to function as an antimetabolite, which is a unique mode of action for a cyanobacterial-derived adaptive substance. Further, because this pathway is absent in mammals, this herbicidal substance may have considerable promise as an environmentally benign agrichemical. All in all, this is an interesting study and certain to generate interest among the community. Below are several comments offered to improve aspects of the manuscript, with point 13 being the one of greatest consequence.

1. Using the term "Sweet" in the title seems superfluous, as the 'sweetness' property is not relevant to the biology being explored. It seems it is being included only because it is a sugar, and it is not clear that this one is actually "sweet". Suggest the authors remove this term from the

title.

2. line 70, consider start sentence with "Because the shikimate pathway is absent in humans and other mammals,"

3. line 96-98 - not quite correct, as these cyanobacteria do have the RiPPs pathway - cite to van der Donk - See PNAS 2010, 107, 10430-35.

3. line 110 – add word as follows: "...first identified natural cyanobacterial antimetabolite"

4. line 141-143 – awkward sentence

5. line 149 – change to "...C2-unit onto a C5-precursor..."

6. line 226 - data addressing "glycogen degradation" seems absent, so if this is from a reference, it should be cited here

7. line 253 – to fully appreciate the functioning of this new antimetabolite, Figure S4 should be in primary manuscript

8. line 284 - does *Gluconobacter oxydans* that Ezaki described as being inhibited by 7dSh have the shikimate pathway? Seems to have the pathway - see *Appl Microbiol Biotech* 2014, 98, 2955 and another several papers pulled up in SciFinder on this topic

9. line 317 – "in the presence"

10. line 328 - not sure macrophages are the best choice of mammalian cell line; authors should briefly explain why they have chosen this cell line for these experiments

11. line 338 – modify to include RiPPs, as in point 3 above

12. line 356 - is 5-deoxy-D-ribose known from any other organism? If so, is its biosynthesis known? Seems to be an intermediate in biosynthesis of fluorinated compounds - see *Chemical Science* 2015 6, 1414-19

13. line 366 – As one of the more substantive comments of this reviewer, how does the producer organism resist the toxicity of its own metabolite (e.g. resistance mechanism)? This would be a really important and interesting dimension of the manuscript, and I encourage the authors to try to understand this and add to this paper.

14. line 391 – I would not expect that "taste" to be a good measure of lack of biological effects, and seems in contrast to the proposed title (sweet natural killing). Further, it was not apparent to this reviewer that the sugar in question, namely 7dSh, was present in these studies of sweetness (reference 47). Authors should clarify this point.

15. line 560 – does this kit measure lactate DH? Maybe stating this would be helpful to the reader

16. Figure 3 - would be nice to see HPLCs of inhibited cultures and 500-fold build up of natural substrate of inhibited enzyme

17. in the SI, it is not clear what compound the data is being provided for (e.g. the isolated natural product or the synthetic compound or data from the literature). For example, it is necessary to present the optical rotation data for the isolated natural product and then give the literature value with the appropriate citation.

Answers to reviewers' comments:

Reviewer #1 (Remarks to the Author):

The authors describe the discovery that 7-deoxy-sedoheptulose (7dSh) is an antimicrobial and herbicidal compound that apparently acts through inhibition of an early step of the shikimate pathway, 3-dehydroquinate synthase (DHQS). The work is well done and generally convincing.

Suggestions and comments

1. Comparison of the activity of 7dSh with that of a known DHQS inhibitor like carbaphosphonate would have been interesting.

Answer: Thank you for the comment. Carbaphosphonates are well investigated in *in vitro* enzyme assays with – at first sight – promising K_i -values in the nanomolar range. However, like other DHQS inhibitors, they do not show sufficient *in vivo* activity (described in literature below). Even though carbaphosphonate causes a minor accumulation of DAH(P), it does not show herbicidal activity (Montchamp, J. L., Piehler, L. T., & Frost, J. W. (1992). Diastereoselection and *in vivo* inhibition of 3-dehydroquinate synthase. *Journal of the American Chemical Society*, 114(12), 4453-4459. doi:10.1021/ja00038a002).

Due to this limitation, together with the fact that inhibitors like carbaphosphonate are not commercially available and chemical synthesis of this compound is very complex, we decided not to test these compounds in our systems.

2. I would prefer a briefer title. "Sweet natural killing" does grab the eye, but is 7dSh even sweet? 7dSh is not new, as it was first reported in 1970. The biological activity that is described is new. "Powerfully" is in the eye of the beholder. Compared to most commercial herbicides, it is not powerful. Glyphosate is not powerful either, as it is a high use rate herbicide compared to many others.

Answer: We thank you for the suggestion and agree to change the title to a briefer one.

In fact, 7dSh is described as sweet and bitter, see:

Lee, C. K., & Birch, G. G. (1976). Structural Functions of Taste in the Sugar Series VII: Taste Properties of Ketoses. *Journal of Pharmaceutical Sciences*, 65(8), 1222-1225. doi: <http://dx.doi.org/10.1002/jps.2600650823>

3. Lines 196-198. The phenomenon could also or partly be due to metabolic degradation or alteration of 7dSh by the cells.

Answer: Thank you, we agree, and have added the comment to the manuscript, see line 194-195

4. Lines 225-227 – I disagree with this analysis. Similar results might be expected with inhibitors of targets in the chloroplast such as phytoene desaturase, HPPD, etc.

Answer: The analysis was performed with the cyanobacterium *Synechocystis*. Cyanobacteria do not have chloroplasts. Extensive studies in our lab concerning the resuscitation of chlorotic *Synechocystis* cells showed, that the first phase of recovery is characterized by the re-establishment of central anabolic reactions and of the basic enzymatic machinery, well before the photosynthetic machinery is re-installed (see Klotz et al., 2016 doi:10.1016/j.cub.2016.08.054 and Spät et al., doi:10.1074/mcp.RA118.000699). This early step of recovery is blocked by 7dSh.

5. Fig. 6c is not referenced or discussed in the text. Please change the Y axis of Fig. 6c to molar measurements instead of peak areas. The amount of shikimate accumulating in glyphosate-treated plants is very large, leading some to think that inhibition of EPSPS leads to deregulation of the shikimate pathway, redirecting carbon from carbon fixation processes. So, do you see similar amounts of carbon accumulating as DAHP as you do as shikimate with glyphosate?

Answer: Thank you for your note. We performed additional experiments to present solid data for quantification of DAHP in *A. variabilis* (see Extended Data Figure 3a). We furthermore replaced the initial Figure 6c. The accumulation of DAHP and Shikimate-3P (S3P) in plants is now depicted in Extended Data Figure 8. Since the amount of accumulating DAHP is depicted for *A. variabilis* in Extended Data Figure 3a, we did not quantify the absolute amount of DAHP or S3P for *A. thaliana*. We here want to show the accumulation of DAHP in 7dSh treated plants without further quantification (Extended Data Figure 8).

6. Petri dish assays of herbicides are often very misleading. Herbicides are used in two main ways – foliar sprays or in soil. Is there significant herbicidal activity when sprayed on plants? If not, is there any activity in seedlings germinating in soil?

Answer: Thank you for your question. In order to address your note, we added further experiments and tested the germination of *A. thaliana* on soil, which is depicted now in Figure 6c.

7. I am not a toxicologist, but was there any effect on THP1 cell growth?

Answer: No, we did not observe any effect on growth, morphology or cell lysis of THP1 cells.

Lines 385-387 – The contradictory results are mainly due to a retracted paper. The toxicologists who I know do not give credence to the reports of carcinogenic properties of glyphosate. See the special issue of *Critical Reviews in Toxicology* (Supplement 1 of volume 46 in 2016) for a full analysis.

Answer: Thank you for your information which guided us to further literature studies. Since it is not our intention to discuss the potential harmful details of glyphosate we have decided to delete the section.

Provide scale bars in Fig. 7b. These micrographs are not very clear. I suggest relegation of Fig. 7 to supporting data.

Answer: Thank you for the suggestion. We added scale bars and relegated the figure to Extended Data Figure 9.

8. Lines 377-379 – How can this statement be made unless these compounds are compared directly in the same bioassay with 7dSh and glyphosate?

Answer: Thank you for pointing to this unclear description. The cited literature did not report potent *in vivo* activity. However, we modified the text for a better understanding (Line 388).

9. Line 389 – The reference does not support the statement. There are papers that show that the half life of AMPA is longer than that of glyphosate in soil. But compared to many other pesticides, the half life of neither is a significant concern, although some have hyped the half life of AMPA.

Answer: Thank you for this information. Since it is not our intention to discuss the properties of glyphosate in detail we deleted the section.

10. Lines 391-392 – I am not clear as to why the taste of 7dSh supports the view that it has no undesirable side effects. Is it sweet? The title of the paper suggest that it is.

Answer: In this paper test persons have tasted 7dSh and described the compound as bitter and sweet. Since test persons obviously did not suffer any harm, this paper supports the statement that 7dSh has no significant undesirable side effects. We could send you the paper if you would like a deeper insight in the work. However, since this statement was also not clear to the other reviewers, we decided to delete the sentence.

Wording suggestions and minor comments:

Line 63 - ..animals, especially mammals. I make this suggestion because humans are animals, as are insects, etc.

Line 70 - ..in arrested growth or...

Line 76 - ..initial commercialization in... It may have been approved earlier than 1974.

Lines 88 & 86 – The target is not very potent, but the antimetabolites are. So, change to: ..Another target of very potent antimetabolites...

Line 121 - ..content apparently peaked..

Line 292 – molar concentration?

Line 294 - ..inhibition of microbes is.. This is not true of glyphosate as a herbicide.

Line 314 – Change obstructed to impaired.

Line 317 – But, was growth inhibited during this time?

Line 324 – To many plant scientists, the seed has germinated when the radicle penetrates the seed coat. After that, what we see is seedling growth. So, I would say:germination and seedling growth process.

Line 379 – activity for what?

Line 387 – ‘Inherited’ is a biological term. Glyphosate has not inherited anything.

Line 392 – Delete pure

Line 646 –Ref. 14 - Do the authors have names?

Answer: Wording suggestions were accepted as recommended, thank you!

Reviewer #2 (Remarks to the Author):

NCOMMS-18-20996-T Reviewer's Comments

This is a generally well-written and comprehensive manuscript. I believe that it can become publishable after the following comments are addressed.

Statistical analysis is particularly weak and must be remedied.

Answer: Thank you for your note. We have remedied and improved the statistical analysis in our work (see Figure 3a, 4b, 5, 6bc, Ext. Data Fig 3a, 5b, 6, 7, 9).

There are more than a few confusing sentences, mostly due to incorrect English usage and grammatical constructions. For example, line 62 'in their application' is awkward, line 67 'in these organisms' is unclear; line 77 'became' should be 'has become'; line 89 'this enzyme' reference is unclear; 103 'collapsing' ?; 141 a compound cannot prefer anything; 224 'reinstall'?; 314 'obstructed?'; etc etc.

Answer: Thank you for your comment. We corrected our sentences according to your suggestions. Exception: "collapsing" – this wording is used in cited literature.

The first paragraph of the introduction is inappropriate since there is no evidence that 7dSh plays any role in microbial communities or ecosystems, only speculation from the authors.

Answer: Thank you for your note. You are correct that it is only speculation. Therefore, we deleted the unnecessary paragraph.

There is evidence that accumulation of shikimate 3-phosphate alone is inhibitory after glyphosate treatment.

Answer: Unfortunately, we are not able to allocate this comment. Do you think we are missing this part in our manuscript?

Line

83: this sentence ignores all other inhibitors of the shikimate pathway, some of which are later referenced by the authors.

Answer: To our knowledge - except for glyphosate - no inhibitor with potent *in vivo* inhibitory activity on enzymes of the shikimate pathway is described in literature. Later referenced compounds showed promising *in vitro* activity, however, failed in *in vivo* application. Glyphosate is the exception.

91: allelochemicals can be inhibitory to many other organisms besides just 'closely related organisms'

Answer: Thank you, we replaced 'closely related' by 'rival' (see line 86)

110: this statement ignores the previously described SF-666B compound.

Answer: SF-666B and 7dSh are the identical compounds. The compound has been isolated before, however, it has never been described as an inhibitor of the shikimate pathway. We are sorry for the misunderstanding.

197: this argument implies irreversible binding, for which there is no evidence. In fact, the yeast results support a reversible binding mechanism. The authors also do not address the possibility of 7dSh catabolism, which is an equally likely explanation.

Answer: Thank you for the note. We have modified the sentence and added the possible metabolic alteration of 7dSh, see line 194-195

206: The slow decrease in photosynthesis could be compared to any number of inhibitors, not just neomycin.

Answer: Yes, any inhibitors that do not directly target the photosynthesis. Photosynthesis inhibitors like DCMU result in an immediate decrease of the photosynthetic activity. Since we used neomycin as a positive control, we suggest to compare the effect of 7dSh to the effect of neomycin.

272: Data have not been analyzed appropriately to make the claim that the delay was 'not less significant'.

Answer: Thank you for the comment. We corrected the sentence (line 269).

282: The referenced work only used glyphosate, not 'other inhibitors of the shikimate pathway'.

Answer: Thank you for the comment. We corrected the sentence (line 278).

311, 313: There are no data presented on seed germination, only seedling growth.

Answer: Thank you for the comment. We corrected the sentence (line 308-310).

315: same as what?

Answer: Thank you for the comment. We modified the sentence for a better understanding (line 312).

316: 'more distinct' is unclear and not quantifiable

Answer: Thank you for the comment. We modified the sentence for a better understanding (line 313).

317: 7 days following what? Which inhibitors? Does this mean they all stopped growing or they reverted to untreated growth rates?

Answer: Thank you for the note. We modified the sentence for a better understanding (line 314-315).

321: de novo was not italicized and was hyphenated on line 226

Answer: Thank you for the comment. We have aligned style of writing (line 330).

323: germination was completed in the presence of both inhibitors, but seedling growth was inhibited.

Thank you for the comment. We corrected the sentence (line 332).

327: THP-1 cells are used mainly as in vitro cancer cell models and monocyte-macrophage differentiation processes, and thus are not the best model to study mammalian toxicity. Detecting cell lysis or observing morphological appearance are neither sensitive nor appropriate measures of toxicity. CHO or other cell lines would be a better choice. 7dSH effects on this one cell line cannot be used to extrapolate to other cell types or mammals.

Thank you for your clarification. We have added several experiments and now tested the effect of 7dSh and glyphosate on other cell lines (see Extended Data Figure 9).

338: considered by whom?

Answer: Thank you for the note. We modified the sentence for a better understanding (line 348-350).

342: this sentence seems entirely speculative. Are there any data?

Answer: We have shown that the cyanobacterial transketolase is able to catalyze the conversion of 5-deoxy-D-ribose to 7dSh. The transketolase is not part of a biosynthetic gene cluster. Therefore, it is very likely that the synthesis of 7dSh (and also of 5-deoxy-D-ribose) is a result of targeted/untargeted enzymatic promiscuity.

361: conceivable but entirely too speculative for this paper

Answer: Thank you for your comment. We agree and deleted the sentence.

362-369: this is nothing but speculation

Answer: Thank you for your note. Yes, the last part of the biological role of 7dSh for the producer strain is speculation (line 374 to 382). Since we would like to address this biological role in the future work we think this speculation is acceptable.

378: sufficient for what?

Answer: Thank you for the note. We modified the sentence for a better understanding (line 389).

380: up-scaled is not a word

Answer: Thank you for the note. We have corrected the wording (line 391).

386: are there references for these recent studies? The possible carcinogenicity of glyphosate is not 'intensely discussed' among knowledgeable herbicide physiologists, since reports of such activity are unreliable, unpublished, or have been retracted from refereed journals. What are 'inherited properties?'

Answer: Thank you for this information. We agree and decided to delete the section.

389: 'High specificity and potent growth inhibition' are in no way related to a lack of undesirable side effects.

391: Cyanide smells like almonds—does that similarly support the expectation of a lack of undesirable side effects?

392: The allelopathy literature is replete with examples of 'pure natural products' that are extremely toxic and bioaccumulate.

Answer: Thank you for your comments. We have deleted the whole section due to entitled reviewer comments.

393-397: The potential to develop this compound as a herbicide is remote. As done here, the addition of 7dSH to liquid growth medium bypasses all plant uptake and translocation requirements, which are not trivial. If this compound is being considered as a soil-applied herbicide (for root uptake), there are a host of complicating factors (formulation, degradation by soil microorganisms, persistence in soil, etc) that would have to be overcome. Foliar applications likewise present even more obstacles. It is enough for the authors to say that it is a potent inhibitor of the shikimate pathway, but extrapolation to commercial herbicide development is beyond the scope and should be deleted.

Answer: Thank you for your note. We have deleted the whole section.

Figures 3 – 6, 7c: none of these data are analyzed correctly. SD bars by themselves cannot be used to make comparisons among treatments. Mean separations and statistical comparisons must be done.

Answer: Thank you for your note. We have remedied and improved the statistical analysis in our work (see Figure 3a, 4b, 5, 6bc, Ext. Data Fig 3a, 5b, 6, 7, 9).

Almost all figures have confusing or awkward axis labels.

Answer: We are sorry to hear about confusing axis labels. We have re-formatted the axis labeling and hope that this is more appealing now

Figure 5: 'Area of MS peak base' is confusing and inappropriate. These values should be compared to standard curves of known standards and expressed in biologically relevant units like umoles per mg of tissue. If the focus is the rate of accumulation or decline, data should be regressed over time with R values, rate constants calculated, and statistically compared.

Answer: Thank you for your note. Comparing the EIC peak areas of certain metabolites is a valuable tool for the relative amount of searched metabolites. Of course this comparison does not give absolute values, however, it allows a relative quantification (more/less in sample A compared to B). Nevertheless, as you suggested, we have now added results from new experiments: As a proof of principle we determined the absolute amount of stated amino acids in Figure 5.

Reviewer #3 (Remarks to the Author):

*The authors present the results of a highly interesting study on a metabolite produced by *Synechococcus elongatus* that inhibits the growth of other cyanobacteria as well as of plants. The authors then isolate this substance and define it as an unusual deoxy-sugar. As such, it is an antimetabolite that interferes with shikimate biosynthesis. By measuring the build-up of unused substrate in inhibitor treated cultures, they deduced the site of inhibition to be 3-dehydroquinate synthase. Based on the structure of the inhibitor and its site of action, it appears to function as an antimetabolite, which is a unique mode of action for a cyanobacterial-derived adaptive substance. Further, because this pathway is absent in mammals, this herbicidal substance may have considerable promise as an environmentally benign agrichemical. All in all, this is an interesting study and certain to generate interest among the community. Below are several comments offered to improve aspects of the manuscript, with point 13 being the one of greatest consequence.*

1. Using the term "Sweet" in the title seems superfluous, as the 'sweetness' property is not relevant to the biology being explored. It seems it is being included only because it is a sugar, and it is not clear that this one is actually "sweet". Suggest the authors remove this term from the title.

Answer: We thank you for the suggestion and agree to change the title to a briefer one.

2. line 70, consider start sentence with *“Because the shikimate pathway is absent in humans and other mammals,”*

Answer: We thank you for the suggestion, we added ‘mammals’ to the sentence, see line 64

3. line 96-98 - *not quite correct, as these cyanobacteria do have the RPPs pathway - cite to van der Donk - See PNAS 2010, 107, 10430-35.*

Answer: We thank you for this interesting literature. The cited work fits perfectly to our estimated biosynthesis model for 7dSh, which might be based on enzymatic promiscuity. We added the literature in line 93-95

3. line 110 – *add word as follows: “...first identified natural cyanobacterial antimetabolite”*

Answer: Thank you for your suggestion. However, we would prefer to not narrow down the statement to cyanobacteria. To our knowledge 7dSh (or SF666B) is the first described natural inhibitor targeting the shikimate pathway and obviously, it is not only produced by cyanobacteria.

4. line 141-143 – awkward sentence

Answer: We modified the sentence for a better understanding (line 138).

5. line 149 – *change to “...C2-unit onto a C5-precursor...”*

Answer: We have accepted the change as recommended, thank you (line 146)

6. line 226 - *data addressing “glycogen degradation” seems absent, so if this is from a reference, it should be cited here*

Answer: Thank you for the comment, we have added the reference (line 225)

7. line 253 – *to fully appreciate the functioning of this new antimetabolite, Figure S4 should be in primary manuscript*

Answer: Thank you for your suggestion. However, since the mode of action is only a suggestion and since we are limited on figures in the main text, we would prefer keeping the figure in the supplements.

8. line 284 - does *Gluconobacter oxydans* that Ezaki described as being inhibited by 7dSh have the shikimate pathway? Seems to have the pathway - see *Appl Microbiol Biotech* 2014, 98, 2955 and another several papers pulled up in SciFinder on this topic

Answer: Thank you for the interesting question. Yes, *G. oxidans* has the shikimate pathway. If you wonder, why we did not observe an activity of 7dSh against *G. oxidans*: We have tested *G. oxidans* on complex media (as described in the original publication). At that time, we did not know about the inhibition of the shikimate pathway. In complex media the end products of the shikimate pathway can be taken up from the media, mitigating the activity of 7dSh. We therefore question the previously described activity in complex media. We can only speculate, that *G. oxidans* may have been cultivated in another medium or that in fact a different *Gluconobacter* strain has been used.

9. line 317 – “in the presence”

Answer: We have accepted the change as recommended, thank you (line 315)

10. line 328 - not sure macrophages are the best choice of mammalian cell line; authors should briefly explain why they have chosen this cell line for these experiments

Answer: We have tested the effect of 7dSh and glyphosate on further cell lines (see Extended Data Figure 9).

11. line 338 – modify to include RiPPs, as in point 3 above

Answer: We have included the RiPPs as recommended, thank you (line 354-356)

12. line 356 - is 5-deoxy-D-ribose known from any other organism? If so, is its biosynthesis known? Seems to be an intermediate in biosynthesis of fluorinated compounds - see *Chemical Science* 2015 6, 1414-19

Answer: Thank you for your literature. However, in cited paper only the fluorinated version of 5-deoxy-D-ribose is mentioned (5-fluoro-5-deoxy-D-ribose). The compound 5-fluoro-5-deoxy-D-ribose seems to be a more frequent natural product (see also DOI: 10.1039/B400754A). To our knowledge 5-deoxy-D-ribose has never been isolated from nature before. The compound is also not described in Dictionary of Natural Products. Elucidation of the biosynthesis of 5-deoxy-D-ribose and 7dSh are part of ongoing work.

13. line 366 – As one of the more substantive comments of this reviewer, how does the producer organism resist the toxicity of its own metabolite (e.g. resistance mechanism)? This would be a really important and interesting dimension of the manuscript, and I encourage the authors to try to understand this and add to this paper.

Answer: Thank you for this interesting question. Since 7dSh is excreted to the media we think of an effective efflux transporter for the compound. Identification of this transporter is ongoing work. However, the producer strain is in fact growth inhibited by high concentrations of 7dSh (data not shown).

14. line 391 – I would not expect that "taste" to be a good measure of lack of biological effects, and seems in contrast to the proposed title (sweet natural killing). Further, it was not apparent to this reviewer that the sugar in question, namely 7dSh, was present in these studies of sweetness (reference 47). Authors should clarify this point.

Answer: Thank you for your note. In this paper test persons have tasted 7dSh and described the compound as bitter and sweet. Since test persons did not suffer any harm, this paper supports the statement that 7dSh has no significant undesirable side effects. We could send you the paper if you would like a deeper insight in the work. However, since this statement was also not clear to the other reviewers we deleted the sentence.

15. line 560 – does this kit measure lactate DH? Maybe stating this would be helpful to the reader

Answer: Yes, it is measuring the lactate DH activity. We modified the sentence for a better understanding (line 597-600).

16. Figure 3 - would be nice to see HPLCs of inhibited cultures and 500-fold build up of natural substrate of inhibited enzyme

Answer: Thank you for your suggestion. The DAHP accumulation in *A. variabilis* as a result of 7dSh treatment can be seen in Figure Extended Data 3ab.

17. in the SI, it is not clear what compound the data is being provided for (e.g. the isolated natural product or the synthetic compound or data from the literature). For example, it is necessary to present the optical rotation data for the isolated natural product and then give the literature value with the appropriate citation.

Answer: Thank you for the entitled comment. Data is provided from the synthetic compounds due to higher amount of substance (we added this note to the legend of Extended Data Table 1). Isolation of high amounts of 7dSh and 5dR from the natural producer strain is very cumbersome and time consuming. Purification of the synthesis products is more easy and reliable. However, as you can see in Figure 2 and Extended Data Figure 10 both isolated and synthesized compounds show identical physicochemical properties.

Concerning the optical rotation data, we have not found a literature value except for the one in the early publications of SF-666B ([https://doi.org/10.1016/S0008-6215\(00\)82545-8](https://doi.org/10.1016/S0008-6215(00)82545-8)). The optical rotation was tested in water and determined to +4°. However, no concentration of the tested compound is mentioned, making it difficult to compare it to our data. By measuring the optical rotation in methanol we observed higher and more clear rotation values. All isolates (*S. elongatus*, *S. setonensis* and TK synthesis) showed positive optical rotation values. Same applies for 5-deoxy-D-ribose.

REVIEWERS' COMMENTS:

Reviewer #1 (Remarks to the Author):

In general, the authors have satisfactorily dealt with my concerns. The addition of Fig. 6c to the paper helps considerably. This helps to take care of a major concern of mine and of Reviewer 2.

Reviewer #2 (Remarks to the Author):

NCOMMS-18-20996A Reviewer's Comments on Author's Rebuttal
November 29, 2018

The authors have successfully addressed my previous comments with the following exceptions:

Line 299: Seedlings were affected by 7dSh, not seeds.

309 and 312: Figure 6a does not show effects of 130 uM 7dSh.

314: what are 'clear' cotyledons?

315: for both inhibitors?

320: but was not visible

323: germination effects were not reported here, only effects on seedling growth; investigated in soil

336ff: I recommend that Figure 9 be included in the manuscript, not in Extended Data

348: surprising to us

389: Figure 11),

425: half-strength Murashige and Skoog (MS) salts basal medium

430: three plates or three wells?

431: what is GS90 soil?; subsequently

433: was planted with single seed; was the pH of 7dSh adjusted?

437: was additional water or inhibitor added to the wells during the 18 day period? Which one?

856: suggest 7dSh reduces the growth... (instead of obstructs)

864: effects on germination were not tested here

Figures

1: move 'b' closer to graph

3a: suggest - - - caption should be 'inoculation OD750'

3b: suggest ordinate label should be 'oxygen evolution...'

5: each bar pair can be labeled on the ordinate as the name of the amino acid, eg, tyrosine [pmol/mg DW] etc (instead of 'amino acids')

6: see line 856 above; 6b: meristems is misspelled; horizontal line above control and 25 uM box/whiskers should extend over both 25 uM boxes

I see a number of typographical and grammatical errors in the Extended Data figure legends that can be corrected with a careful proofreading.

Reviewer #3 (Remarks to the Author):

The authors have carefully considered this reviewer's previous comments and concerns, and have provided modifications or expansions to the manuscript that are completely satisfactory. This reviewer has no new concerns or questions regarding the manuscript.

William Gerwick

Answers to reviewers' comments:

Reviewer #1 (Remarks to the Author):

In general, the authors have satisfactorily dealt with my concerns. The addition of Fig. 6c to the paper helps considerably. This helps to take care of a major concern of mine and of Reviewer 2.

Answer: Thank you very much for your help to improve our work.

Reviewer #2 (Remarks to the Author):

The authors have successfully addressed my previous comments with the following exceptions:

Line 299: Seedlings were affected by 7dSh, not seeds.

309 and 312: Figure 6a does not show effects of 130 μ M 7dSh.

314: what are 'clear' cotyledons?

315: for both inhibitors?

320: but was not visible

323: germination effects were not reported here, only effects on seedling growth; investigated in soil

336ff: I recommend that Figure 9 be included in the manuscript, not in Extended Data

348: surprising to us

389: Figure 11),

425: half-strength Murashige and Skoog (MS) salts basal medium

430: three plates or three wells?

431: what is GS90 soil?; subsequently

433: was planted with single seed; was the pH of 7dSh adjusted?

437: was additional water or inhibitor added to the wells during the 18 day period? Which one?

856: suggest 7dSh reduces the growth... (instead of obstructs)

864: effects on germination were not tested here

Figures

1: move 'b' closer to graph

3a: suggest - - - caption should be 'inoculation OD750'

3b: suggest ordinate label should be 'oxygen evolution....'

5: each bar pair can be labeled on the ordinate as the name of the amino acid, eg, tyrosine [pmol/mg DW] etc (instead of 'amino acids')

6: see line 856 above; 6b: meristems is misspelled; horizontal line above control and 25 μ M box/whiskers

should extend over both 25 μ M boxes

I see a number of typographical and grammatical errors in the Extended Data figure legends that can be corrected with a careful proofreading.

Answer: Thank you for your revision. We have corrected our manuscript as suggested for a better understanding.

Comment to line 315: Yes, we did not observe a change for both inhibitors.

Comment to line 336ff: We have moved the figure to Supporting Information due to advice of another Reviewer. We will ask the editor for his opinion.

Comment to line 431: GS90 (pH value: 5.8, salt content: 1.5 g/L) is a standard soil used in the laboratories of the ZMBP Tübingen. It is delivered from Patzer GmbH (email: info@einheitserde.de). For further details, we can send you the product data sheet (00800 CL Ton Kokos).

Comment to line 433: No, since 7dSh does not change the pH value (other than the acid glyphosate).

Comment to line 437: No, we neither added water nor an inhibitor.

Reviewer #3 (Remarks to the Author):

The authors have carefully considered this reviewer's previous comments and concerns, and have provided modifications or expansions to the manuscript that are completely satisfactory. This reviewer has no new concerns or questions regarding the manuscript.

Answer: Thank you very much for your help to improve our work.